# Transfer Learning with Affine Model Transformation

**Shunya Minami**
The Institute of Statistical Mathematics
mshunya@ism.ac.jp

**Kenji Fukumizu**
The Institute of Statistical Mathematics
fukumizu@ism.ac.jp

**Yoshihiro Hayashi**
The Institute of Statistical Mathematics
yhayashi@ism.ac.jp

**Ryo Yoshida**
The Institute of Statistical Mathematics
yoshidar@ism.ac.jp

## Abstract

Supervised transfer learning has received considerable attention due to its potential to boost the predictive power of machine learning in scenarios where data are scarce. Generally, a given set of source models and a dataset from a target domain are used to adapt the pre-trained models to a target domain by statistically learning domain shift and domain-specific factors. While such procedurally and intuitively plausible methods have achieved great success in a wide range of real-world applications, the lack of a theoretical basis hinders further methodological development. This paper presents a general class of transfer learning regression called affine model transfer, following the principle of expected-square loss minimization. It is shown that the affine model transfer broadly encompasses various existing methods, including the most common procedure based on neural feature extractors. Furthermore, the current paper clarifies theoretical properties of the affine model transfer such as generalization error and excess risk. Through several case studies, we demonstrate the practical benefits of modeling and estimating inter-domain commonality and domain-specific factors separately with the affine-type transfer models.

## 1 Introduction

Transfer learning (TL) is a methodology to improve the predictive performance of machine learning in a target domain with limited data by reusing knowledge gained from training in related source domains. Its great potential has been demonstrated in various real-world problems, including computer vision [1, 2], natural language processing [3, 4], biology [5], and materials science [6, 7, 8]. Notably, most of the outstanding successes of TL to date have relied on the feature extraction ability of deep neural networks. For example, a conventional method reuses feature representations encoded in an intermediate layer of a pre-trained model as an input for the target task or uses samples from the target domain to fine-tune the parameters of the pre-trained source model [9]. While such methods are operationally plausible and intuitive, they lack methodological principles and remain theoretically unexplored in terms of their learning capability for limited data. This study develops a principled methodology generally applicable to various kinds of TL.

In this study, we focus on supervised TL settings. In particular, we deal with settings where, given feature representations obtained from training in the source domain, we use samples from the target domain to model and estimate the domain shift to the target. This procedure is called hypothesis transfer learning (HTL); several methods have been proposed, such as using a linear transformation function [10, 11] and considering a general class of continuous transformation functions [12]. If the transformation function appropriately captures the functional relationship between the source and target domains, only the domain-specific factors need to be additionally learned, which can be done efficiently even with a limited sample size. In other words, the performance of HTL depends strongly

37th Conference on Neural Information Processing Systems (NeurIPS 2023).

on whether the transformation function appropriately represents the cross-domain shift. However, the general methodology for modeling and estimating such domain shifts has been less studied.

This study derives a theoretically optimal class of supervised TL that minimizes the expected $\ell_2$ loss function of the HTL. The resulting function class takes the form of an affine coupling $g_1(\boldsymbol{f_s}) + g_2(\boldsymbol{f_s}) \cdot g_3(\boldsymbol{x})$ of three functions $g_1, g_2$ and $g_3$, where the shift from a given source feature $f_s$ to the target domain is represented by the functions $g_1$ and $g_2$, and the domain-specific factors are represented by $g_3(\boldsymbol{x})$ for any given input $\boldsymbol{x}$. These functions can be estimated simultaneously using conventional supervised learning algorithms such as kernel methods or deep neural networks. Hereafter, we refer to this framework as the *affine model transfer*. As described later, we can formulate a wide variety of TL algorithms within the affine model transfer, including the widely used neural feature extractors, offset and scale HTLs [10, 11, 12], and Bayesian TL [13]. We clarify theoretical properties of the affine model transfer such as generalization error and excess risk.

To summarize, the contributions of our study are as follows:

- The affine model transfer is proposed to adapt source features to the target domain by separately estimating cross-domain shift and domain-specific factors.
- The affine form is derived theoretically as an optimal class based on the squared loss for the target task.
- The affine model transfer encompasses several existing TL methods, including neural feature extraction. It can work with any type of source model, including non-machine learning models such as physical models as well as multiple source models.
- For each of the three functions $g_1$, $g_2$, and $g_3$, we provide an efficient and stable estimation algorithm when modeled using the kernel method.
- Two theoretical properties of the affine transfer model are shown: the generalization and the excess risk bound.

With several applications, we compare the affine model transfer with other TL algorithms, discuss its strengths, and demonstrate the advantage of being able to estimate cross-domain shifts and domain-specific factors separately.

## 2 Transfer Learning via Transformation Function

### 2.1 Affine Model Transfer

This study considers regression problems with squared loss. We assume that the output of the target domain $y \in \mathcal{Y} \subset \mathbb{R}$ follows $y = f_t(\boldsymbol{x}) + \epsilon$, where $f_t : \mathcal{X} \to \mathbb{R}$ is the true model on the target domain, and the observation noise $\epsilon$ has mean zero and variance $\sigma^2$. We are given $n$ samples $\{(\boldsymbol{x}_i, y_i)\}_{i=1}^n \in (\mathcal{X} \times \mathcal{Y})^n$ from the target domain and the feature representation $\boldsymbol{f_s}(\boldsymbol{x}) \in \mathcal{F}_s$ from one or more source domains. Typically, $\boldsymbol{f_s}$ is given as a vector, including the output of the source models, observed data in the source domains, or learned features in a pre-trained model, but it can also be a non-vector feature such as a tensor, graph, or text. Hereafter, $\boldsymbol{f_s}$ is referred to as the source features.

In this paper, we focus on transfer learning with variable transformations as proposed in [12]. For an illustration of the concept, consider the case where there exists a relationship between the true functions $f_s^*(\boldsymbol{x}) \in \mathbb{R}$ and $f_t^*(\boldsymbol{x}) \in \mathbb{R}$ such that $f_t^*(\boldsymbol{x}) = f_s^*(\boldsymbol{x}) + \boldsymbol{x}^\top \boldsymbol{\theta}^*, \boldsymbol{x} \in \mathbb{R}^d$ with an unknown parameter $\boldsymbol{\theta}^* \in \mathbb{R}^d$. If $f_s^*$ is non-smooth, a large number of training samples is needed to learn $f_t^*$ directly. However, since the difference $f_t^* - f_s^*$ is a linear function with respect to the unknown $\boldsymbol{\theta}^*$, it can be learned with fewer samples if prior information about $f_s^*$ is available. For example, a target model can be obtained by adding $f_s$ to the model $g$ trained for the intermediate variable $z = y - f_s(\boldsymbol{x})$.

The following is a slight generalization of TL procedure provided in [12]:

1. With the source features, perform a variable transformation of the observed outputs as $z_i = \phi(y_i, \boldsymbol{f_s}(\boldsymbol{x}_i))$, using the data transformation function $\phi : \mathcal{Y} \times \mathcal{F}_s \to \mathbb{R}$.
2. Train an intermediate model $\hat{g}(\boldsymbol{x})$ using the transformed sample set $\{(\boldsymbol{x}_i, z_i)\}_{i=1}^n$ to predict the transformed output $z$ for any given $\boldsymbol{x}$.
3. Obtain a target model $\hat{f}_t(\boldsymbol{x}) = \psi(\hat{g}(\boldsymbol{x}), \boldsymbol{f_s}(\boldsymbol{x}))$ using the model transformation function $\psi : \mathbb{R} \times \mathcal{F}_s \to \mathcal{Y}$ that combines $\hat{g}$ and $\boldsymbol{f_s}$ to define a predictor.

In particular, [12] considers the case where the model transformation function is equal to the inverse of the data transformation function. We consider a more general case that eliminates this constraint.

The objective of step 1 is to identify a transformation $\phi$ that maps the output variable $y$ to the intermediate variable $z$, making the variable suitable for learning. In step 2, a predictive model for $z$ is constructed. Since the data is limited in many TL setups, a simple model, such as a linear model, should be used as $g$. Step 3 is to transform the intermediate model $g$ into a predictive model $f_t$ for the original output $y$.

This class of TL includes several approaches proposed in previous studies. For example, [10, 11] proposed a learning algorithm consisting of linear data transformation and linear model transformation: $\phi = y - \langle \boldsymbol{\theta}, \boldsymbol{f_s} \rangle$ and $\psi = g(\boldsymbol{x}) + \langle \boldsymbol{\theta}, \boldsymbol{f_s} \rangle$ with pre-defined weights $\boldsymbol{\theta}$. In this case, factors unexplained by the linear combination of source features are learned with $g$, and the target output is predicted additively with the common factor $\langle \boldsymbol{\theta}, \boldsymbol{f_s} \rangle$ and the additionally learned $g$. In [13], it is shown that a type of Bayesian TL is equivalent to using the following transformation functions; for $\mathcal{F}_s \subset \mathbb{R}$, $\phi = (y - \tau f_s)/(1 - \tau)$ and $\psi = \rho g(\boldsymbol{x}) + (1-\rho)f_s$ with two varying hyperparameters $\tau < 1$ and $0 \le \rho \le 1$. This includes TL using density ratio estimation [14] and neural network-based fine-tuning as special cases when the two hyperparameters belong to specific regions.

The performance of this TL strongly depends on the design of the two transformation functions $\phi$ and $\psi$. In the sequel, we theoretically derive the optimal form of transformation functions under the squared loss scenario. For simplicity, we denote the transformation functions as $\phi_{f_s}(\cdot) = \phi(\cdot, \boldsymbol{f_s})$ on $\mathcal{Y}$ and $\psi_{f_s}(\cdot) = \psi(\cdot, \boldsymbol{f_s})$ on $\mathbb{R}$. To derive the optimal class of $\phi$ and $\psi$, note first that the TL procedure described above can be formulated in population as solving two successive least square problems;

$$(i) \ g^* := \arg\min_g \mathbb{E}_{p_t} \|\phi_{f_s}(Y) - g(\boldsymbol{X})\|^2, \qquad (ii) \ \min_\psi \mathbb{E}_{p_t} \|Y - \psi_{f_s}(g^*(\boldsymbol{X}))\|^2.$$

Since the regression function that minimizes the mean squared error is the conditional mean, the first problem is solved by $g_\phi^*(\boldsymbol{x}) = \mathbb{E}[\phi_{f_s}(Y)|\boldsymbol{X} = \boldsymbol{x}]$, which depends on $\phi$. We can thus consider the optimal transformation functions $\phi$ and $\psi$ by the following minimization:

$$\min_{\psi,\phi} \mathbb{E}_{p_t} \|Y - \psi_{f_s}(g_\phi^*(\boldsymbol{X}))\|^2. \tag{1}$$

It is easy to see that Eq. (1) is equivalent to the following consistency condition:

$$\psi_{f_s}\big(g_\phi^*(\boldsymbol{x})\big) = \mathbb{E}_{p_t}[Y|\boldsymbol{X} = \boldsymbol{x}].$$

From the above observation, we make three assumptions to derive the optimal form of $\psi$ and $\phi$:

**Assumption 2.1** (Differentiability). The data transformation function $\phi$ is differentiable with respect to the first argument.

**Assumption 2.2** (Invertibility). The model transformation function $\psi$ is invertible with respect to the first argument, i.e., its inverse $\psi_{f_s}^{-1}$ exists.

**Assumption 2.3** (Consistency). For any distribution on the target domain $p_t(\boldsymbol{x}, y)$, and for all $\boldsymbol{x} \in \mathcal{X}$,

$$\psi_{f_s}(g^*(\boldsymbol{x})) = \mathbb{E}_{p_t}[Y|\boldsymbol{X} = \boldsymbol{x}],$$

where $g^*(\boldsymbol{x}) = \mathbb{E}_{p_t}[\phi_{f_s}(Y)|\boldsymbol{X} = \boldsymbol{x}]$.

Assumption 2.2 is commonly used in most existing HTL settings, such as [10] and [12]. It assumes a one-to-one correspondence between the predictive value $\hat{f}_t(\boldsymbol{x})$ and the output of the intermediate model $\hat{g}(\boldsymbol{x})$. If this assumption does not hold, then multiple values of $\hat{g}$ correspond to the same predicted value $\hat{f}_t$, which is unnatural. Note that Assumption 2.3 corresponds to the unbiased condition of [12].

We now derive the properties that the optimal transformation functions must satisfy.

**Theorem 2.4.** *Under Assumptions 2.1-2.3, the transformation functions $\phi$ and $\psi$ satisfy the following two properties:*

$$(i) \ \ \psi_{f_s}^{-1} = \phi_{f_s}. \qquad\qquad (ii) \ \ \psi_{f_s}(g) = g_1(\boldsymbol{f_s}) + g_2(\boldsymbol{f_s}) \cdot g,$$

*where $g_1$ and $g_2$ are some functions.*

The proof is given in Section D.1 in Supplementary Material. Despite not initially assuming that the two transformation functions are inverses, Theorem 2.4 implies they must indeed be inverses. Furthermore, the mean squared error is minimized when the data and model transformation functions are given by an affine transformation and its inverse, respectively. In summary, under the expected squared loss minimization with the HTL procedure, the optimal class for HTL model is expressed as follows:

$$\mathcal{H} = \big\{ \boldsymbol{x} \mapsto g_1(\boldsymbol{f_s}) + g_2(\boldsymbol{f_s}) g_3(\boldsymbol{x}) \mid g_j \in \mathcal{G}_j, j = 1, 2, 3 \big\},$$

where $\mathcal{G}_1, \mathcal{G}_2$ and $\mathcal{G}_3$ are the arbitrarily function classes. Here, each of $g_1$ and $g_2$ is modeled as a function of $\boldsymbol{f_s}$ that represents common factors across the source and target domains. $g_3$ is modeled as a function of $\boldsymbol{x}$, in order to capture the domain-specific factors unexplainable by the source features.

We have derived the optimal form of the transformation functions when the squared loss is employed. Even for general convex loss functions, (i) of Theorem 2.4 still holds. However, (ii) of Theorem 2.4 does not generally hold because the optimal transformation function depends on the loss function. Extensions to other losses are briefly discussed in Section A.1, but the establishment of a complete theory is a future work.

Here, the affine transformation is found to be optimal in terms of minimizing the mean squared error. We can also derive the same optimal function by minimizing the upper bound of the estimation error in the HTL procedure, as discussed in Section A.2.

One of key principles for the design of $g_1$, $g_2$, and $g_3$ is interpretability. In our model, $g_1$ and $g_2$ primarily facilitate knowledge transfer, while the estimated $g_3$ is used to gain insight on domain-specific factors. For instance, in order to infer cross-domain differences, we could design $g_1$ and $g_2$ by the conventional neural feature extraction, and a simple, highly interpretable model such as a linear model could be used for $g_3$. Thus, observing the estimated regression coefficients in $g_3$, one can statistically infer which features of $\boldsymbol{x}$ are related to inter-domain differences. This advantage of the proposed method is demonstrated in Section 5.2 and Section B.3.

## 2.2   Relation to Existing Methods

The affine model transfer encompasses some existing TL procedures. For example, by setting $g_1(\boldsymbol{f_s}) = 0$ and $g_2(\boldsymbol{f_s}) = 1$, the prediction model is estimated without using the source features, which corresponds to an ordinary direct learning, i.e., a learning scheme without transfer. Furthermore, various kinds of HTLs can be formulated by imposing constraints on $g_1$ and $g_2$. In prior work, [10] employs a two-step procedure where the source features are combined with pre-defined weights, and then the auxiliary model is additionally learned for the residuals unexplainable by the source features. The affine model transfer can represent this HTL as a special case by setting $g_2 = 1$. [12] uses the transformed output $z_i = y_i / f_s$ with the output value $f_s \in \mathbb{R}$ of a source model, and this cross-domain shift is then regressed onto $\boldsymbol{x}$ using a target dataset. This HTL corresponds to $g_1 = 0$ and $g_2 = f_s$.

When a pre-trained source model is provided as a neural network, TL is usually performed with the intermediate layer as input to the model in the target domain. This is called a feature extractor or frozen featurizer and has been experimentally and theoretically proven to have strong transfer capability as the de facto standard for TL [9, 15]. The affine model transfer encompasses the neural feature extraction as a special subclass, which is equivalent to setting $g_2(\boldsymbol{f_s}) g_3(\boldsymbol{x}) = 0$. A performance comparison of the affine model transfer with the neural feature extraction is presented in Section 5 and Section B.2. The relationships between these existing methods and the affine model transfer are illustrated in Figure 1 and Figure S.1

The affine model transfer can also be interpreted as generalizing the feature extraction by adding a product term $g_2(\boldsymbol{f_s}) g_3(\boldsymbol{x})$. This additional term allows for the inclusion of unknown factors in the transferred model that are unexplainable by source features alone. Furthermore, this encourages the avoidance of a negative transfer, a phenomenon where prior learning experiences interfere with training in a new task. The usual TL based only on $\boldsymbol{f_s}$ attempts to explain and predict the data generation process in the target domain using only the source features. However, in the presence of domain-specific factors, a negative transfer can occur owing to a lack of descriptive power. The additional term compensates for this shortcoming. The comparison of behavior for the case with the non-relative source features is described in Section 5.1.

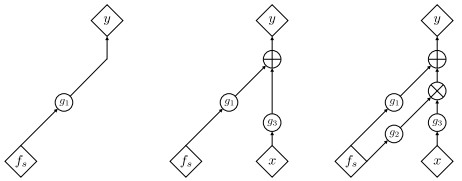

Figure 1: Architectures of (a) feature extraction, (b) HTL in [10], and (c) affine model transfer.

**Algorithm 1** Block relaxation algorithm [19].

> **Initialize:** $\boldsymbol{a}_0, \boldsymbol{b}_0 \neq \boldsymbol{0}, \boldsymbol{c}_0 \neq \boldsymbol{0}$
> **repeat**
>   $\boldsymbol{a}_{t+1} = \arg\min_{\boldsymbol{a}} F(\boldsymbol{a},\ \boldsymbol{b}_t,\ \boldsymbol{c}_t)$
>   $\boldsymbol{b}_{t+1} = \arg\min_{\boldsymbol{b}} F(\boldsymbol{a}_{t+1},\ \boldsymbol{b},\ \boldsymbol{c}_t)$
>   $\boldsymbol{c}_{t+1} = \arg\min_{\boldsymbol{c}} F(\boldsymbol{a}_{t+1},\ \boldsymbol{b}_{t+1},\ \boldsymbol{c})$
> **until** convergence

The affine model transfer can be naturally expressed as an architecture of network networks. This architecture, called affine coupling layers, is widely used for invertible neural networks in flow-based generative modeling [16, 17]. Neural networks based on affine coupling layers have been proven to have universal approximation ability [18]. This implies that the affine transfer model has the potential to represent a wide range of function classes, despite its simple architecture based on the affine coupling of three functions.

## 3 Modeling and Estimation

In this section, we focus on using kernel methods for the affine transfer model and provide the estimation algorithm. Let $\mathcal{H}_1, \mathcal{H}_2$ and $\mathcal{H}_3$ be reproducing kernel Hilbert spaces (RKHSs) with positive-definite kernels $k_1, k_2$ and $k_3$, which define the feature mappings $\Phi_1 : \mathcal{F}_s \to \mathcal{H}_1, \Phi_2 : \mathcal{F}_s \to \mathcal{H}_2$ and $\Phi_3 : \mathcal{X} \to \mathcal{H}_3$, respectively. Denote $\Phi_{1,i} = \Phi_1(\boldsymbol{f_s}(\boldsymbol{x}_i)), \Phi_{2,i} = \Phi_2(\boldsymbol{f_s}(\boldsymbol{x}_i)), \Phi_{3,i} = \Phi_3(\boldsymbol{x}_i)$. For the proposed model class, the $\ell_2$-regularized empirical risk with the squared loss is given as follows:

$$F_{\alpha,\beta,\gamma} = \frac{1}{n}\sum_{i=1}^{n} \left\{ y_i - \langle \alpha, \Phi_{1,i} \rangle - \langle \beta, \Phi_{2,i} \rangle \langle \gamma, \Phi_{3,i} \rangle \right\}^2 + \lambda_1 \|\alpha\|_{\mathcal{H}_1}^2 + \lambda_2 \|\beta\|_{\mathcal{H}_2}^2 + \lambda_3 \|\gamma\|_{\mathcal{H}_3}^2, \quad (2)$$

where $\lambda_1, \lambda_2, \lambda_3 \geq 0$ are hyperparameters for the regularization. According to the representer theorem, the minimizer of $F_{\alpha,\beta,\gamma}$ with respect to the parameters $\alpha \in \mathcal{H}_1$, $\beta \in \mathcal{H}_2$, and $\gamma \in \mathcal{H}_3$ reduces to $\alpha = \sum_{i=1}^{n} a_i \Phi_{1,i}, \beta = \sum_{i=1}^{n} b_i \Phi_{2,i}, \gamma = \sum_{i=1}^{n} c_i \Phi_{3,i}$, with the $n$-dimensional unknown parameter vectors $\boldsymbol{a}, \boldsymbol{b}, \boldsymbol{c} \in \mathbb{R}^n$. Substituting this expression into Eq. (2), we obtain the objective function as

$$F_{\alpha,\beta,\gamma} = \frac{1}{n} \|y - K_1 \boldsymbol{a} - (K_2 \boldsymbol{b}) \circ (K_3 \boldsymbol{c})\|_2^2 + \lambda_1 \boldsymbol{a}^\top K_1 \boldsymbol{a} + \lambda_2 \boldsymbol{b}^\top K_2 \boldsymbol{b} + \lambda_3 \boldsymbol{c}^\top K_3 \boldsymbol{c} =: F(\boldsymbol{a}, \boldsymbol{b}, \boldsymbol{c}). \quad (3)$$

Here, the symbol $\circ$ denotes the Hadamard product. $K_I$ is the Gram matrix associated with the kernel $k_I$ for $I \in \{1, 2, 3\}$. $k_I^{(i)} = [k_I(\boldsymbol{x}_i, \boldsymbol{x}_1) \cdots k_I(\boldsymbol{x}_i, \boldsymbol{x}_n)]^\top$ denotes the $i$-th column of the Gram matrix. The $n \times n$ matrix $M^{(i)}$ is given by the tensor product $M^{(i)} = k_2^{(i)} \otimes k_3^{(i)}$ of $k_2^{(i)}$ and $k_3^{(i)}$.

Because the model is linear with respect to parameter $a$ and bilinear for $b$ and $c$, the optimization of Eq. (3) can be solved using well-established techniques for the low-rank tensor regression. In this study, we use the block relaxation algorithm [19] as described in Algorithm 1. It updates $a, b$, and $c$ by repeatedly fixing two of the three parameters and minimizing the objective function for the remaining one. Fixing two parameters, the resulting subproblem can be solved analytically because the objective function is expressed in a quadratic form for the remaining parameter.

Algorithm 1 can be regarded as repeating the HTL procedure introduced in Section 2.1; alternately estimates the parameters $(\boldsymbol{a}, \boldsymbol{b})$ of the transformation function and the parameters $\boldsymbol{c}$ of the model for the given transformed data $\{(\boldsymbol{x}_i, z_i)\}_{i=1}^n$. The function $F$ in Algorithm 1 is not jointly convex in general. However, when employing methods like kernel methods or generalized linear models, and fixing two parameters, $F$ exhibits convexity with respect to the remaining parameter. According to [19], when each sub-minimization problem is convex, Algorithm 1 is guaranteed to converge to a stationary point. Furthermore, [19] showed that consistency and asymptotic normality hold for the alternating minimization algorithm.

## 4 Theoretical Results

In this section, we present two theoretical properties, the generalization bound and excess risk bound.

Let $(\mathcal{Z}, P)$ be an arbitrary probability space, and $\{z_i\}_{i=1}^n$ be independent random variables with distribution $P$. For a function $f : \mathcal{Z} \to \mathbb{R}$, let the expectation of $f$ with respect to $P$ and its empirical counterpart denote respectively by $Pf = \mathbb{E}_P f(z), P_n f = \frac{1}{n} \sum_{i=1}^n f(z_i)$. We use a non-negative loss $\ell(y, y')$ such that it is bounded from above by $L > 0$ and for any fixed $y' \in \mathcal{Y}$, $y \mapsto \ell(y, y')$ is $\mu_\ell$-Lipschitz for some $\mu_\ell > 0$.

Recall that the function class proposed in this work is

$$\mathcal{H} = \big\{ x \mapsto g_1(\boldsymbol{f_s}) + g_2(\boldsymbol{f_s}) g_3(\boldsymbol{x}) \mid g_j \in \mathcal{G}, j = 1, 2, 3 \big\}.$$

In particular, the following discussion in this section assumes that $g_1, g_2$, and $g_3$ are represented by linear functions on the RKHSs.

## 4.1 Generalization Bound

The optimization problem is expressed as follows:

$$\min_{\alpha, \beta, \gamma} P_n \ell\big(y, \langle \alpha, \Phi_1 \rangle_{\mathcal{H}_1} + \langle \beta, \Phi_2 \rangle_{\mathcal{H}_2} \langle \gamma, \Phi_3 \rangle_{\mathcal{H}_3}\big) + \lambda_\alpha \|\alpha\|_{\mathcal{H}_1}^2 + \lambda_\beta \|\beta\|_{\mathcal{H}_2}^2 + \lambda_\gamma \|\gamma\|_{\mathcal{H}_3}^2, \quad (4)$$

where $\Phi_1 = \Phi_1(\boldsymbol{f_s}(\boldsymbol{x})), \Phi_2 = \Phi_2(\boldsymbol{f_s}(\boldsymbol{x}))$ and $\Phi_3 = \Phi_3(\boldsymbol{x})$ denote the feature maps. Without loss of generality, it is assumed that $\|\Phi_i\|_{\mathcal{H}_i}^2 \leq 1$ $(i = 1, 2, 3)$ and $\lambda_\alpha, \lambda_\beta, \lambda_\gamma > 0$. Hereafter, we will omit the suffixes $\mathcal{H}_i$ in the norms if there is no confusion.

Let $(\hat{\alpha}, \hat{\beta}, \hat{\gamma})$ be a solution of Eq. (4), and denote the corresponding function in $\mathcal{H}$ as $\hat{h}$. For any $\alpha$, we have

$$\lambda_\alpha \|\hat{\alpha}\|^2 \leq P_n \ell(y, \langle \hat{\alpha}, \Phi_1 \rangle + \langle \hat{\beta}, \Phi_2 \rangle \langle \hat{\gamma}, \Phi_3 \rangle) + \lambda_\alpha \|\hat{\alpha}\|^2 + \lambda_\beta \|\hat{\beta}\|^2 + \lambda_\gamma \|\hat{\gamma}\|^2 \leq P_n \ell(y, \langle \alpha, \Phi_1 \rangle) + \lambda_\alpha \|\alpha\|^2,$$

where we use the fact that $\ell(\cdot, \cdot)$ and $\| \cdot \|$ are non-negative, and $(\hat{\alpha}, \hat{\beta}, \hat{\gamma})$ is the minimizer of Eq. (4). Denoting $\hat{R}_s = \inf_\alpha \{ P_n \ell(y, \langle \alpha, \Phi_1 \rangle) + \lambda_\alpha \|\alpha\|^2 \}$, we obtain $\|\hat{\alpha}\|^2 \leq \lambda_\alpha^{-1} \hat{R}_s$. Because the same inequality holds for $\lambda_\beta \|\hat{\beta}\|^2, \lambda_\gamma \|\hat{\gamma}\|^2$ and $P_n \ell(y, \hat{h})$, we have $\|\hat{\beta}\|^2 \leq \lambda_\beta^{-1} \hat{R}_s, \|\hat{\gamma}\|^2 \leq \lambda_\gamma^{-1} \hat{R}_s$ and $P_n \ell(y, \hat{h}) \leq \hat{R}_s$. Moreover, we have $P\ell(y, \hat{h}) = \mathbb{E} P_n \ell(y, \hat{h}) \leq \mathbb{E} \hat{R}_s$. Therefore, it is sufficient to consider the following hypothesis class $\tilde{\mathcal{H}}$ and loss class $\mathcal{L}$:

$$\tilde{\mathcal{H}} = \big\{ x \mapsto \langle \alpha, \Phi_1 \rangle + \langle \beta, \Phi_2 \rangle \langle \gamma, \Phi_3 \rangle \,\big|\, \|\alpha\|^2 \leq \lambda_\alpha^{-1} \hat{R}_s, \|\beta\|^2 \leq \lambda_\beta^{-1} \hat{R}_s, \|\gamma\|^2 \leq \lambda_\gamma^{-1} \hat{R}_s, P\ell(y, h) \leq \mathbb{E} \hat{R}_s \big\},$$

$$\mathcal{L} = \big\{ (x, y) \mapsto \ell(y, h) \mid h \in \tilde{\mathcal{H}} \big\}.$$

Here, we show the generalization bound of the proposed model class. The following theorem is based on [11], showing that the difference between the generalization error and empirical error can be bounded using the magnitude of the relevance of the domains.

**Theorem 4.1.** *There exists a constant $C$ depending only on $\lambda_\alpha, \lambda_\beta, \lambda_\gamma$ and $L$ such that, for any $\eta > 0$ and $h \in \tilde{\mathcal{H}}$, with probability at least $1 - e^{-\eta}$,*

$$P\ell(y, h) - P_n \ell(y, h) = \tilde{O}\bigg( \bigg( \sqrt{\frac{R_s}{n}} + \frac{\mu_\ell^2 C^2 + \sqrt{\eta}}{n} \bigg)(C + \sqrt{\eta}) + \frac{C^2 + \eta}{n} \bigg),$$

*where $R_s = \inf_\alpha \{ P\ell(y, \langle \alpha, \Phi_1 \rangle) + \lambda_\alpha \|\alpha\|^2 \}$.*

Because $\Phi_1$ is the feature map from the source feature space $\mathcal{F}_s$ into the RKHS $\mathcal{H}_1$, $R_s$ corresponds to the true risk of training in the target domain using only the source features $f_s$. If this is sufficiently small, e.g., $R_s = \tilde{O}(n^{-1/2})$, the convergence rate indicated by Theorem 4.1 becomes $n^{-1}$, which is an improvement over the naive convergence rate $n^{-1/2}$. This means that if the source task yields feature representations strongly related to the target domain, training in the target domain is accelerated. Theorem 4.1 measures this cross-domain relation using the metric $R_s$.

Theorem 4.1 is based on Theorem 11 of [11] in which the function class $g_1 + g_3$ is considered. Our work differs in the following two points: the source features are modeled not only additively but also multiplicatively, i.e., we consider the function class $g_1 + g_2 \cdot g_3$, and we also consider the estimation of the parameters for the source feature combination, i.e., the parameters of the functions $g_1$ and $g_2$. In particular, the latter affects the resulting rate. With fixed the source combination parameters, the resulting rate improves only up to $n^{-3/4}$. The details are discussed in Section D.2.

## 4.2 Excess Risk Bound

Here, we analyze the excess risk, which is the difference between the risk of the estimated function and the smallest possible risk within the function class.

Recall that we consider the functions $g_1, g_2$ and $g_3$ to be elements of the RKHSs $\mathcal{H}_1, \mathcal{H}_2$ and $\mathcal{H}_3$ with kernels $k_1, k_2$ and $k_3$, respectively. Define the kernel $k^{(1)} = k_1$, $k^{(2)} = k_2 \cdot k_3$ and $k = k^{(1)} + k^{(2)}$. Let $\mathcal{H}^{(1)}, \mathcal{H}^{(2)}$ and $\mathcal{H}$ be the RKHS with $k^{(1)}, k^{(2)}$ and $k$ respectively. For $m = 1, 2$, consider the normalized Gram matrix $K^{(m)} = \frac{1}{n}(k^{(m)}(\boldsymbol{x}_i, \boldsymbol{x}_j))_{i,j=1,...,n}$ and its eigenvalues $(\hat{\lambda}_i^{(m)})_{i=1}^n$, arranged in a nonincreasing order.

We make the following additional assumptions:

**Assumption 4.2.** There exists $h^* \in \mathcal{H}$ and $h^{(m)*} \in \mathcal{H}^{(m)}$ $(m = 1, 2)$ such that $P(y - h^*(\boldsymbol{x}))^2 = \inf_{h \in \mathcal{H}} P(y - h(\boldsymbol{x}))^2$ and $P(y - h^{(m)*}(\boldsymbol{x}))^2 = \inf_{h \in \mathcal{H}^{(m)}} P(y - h(\boldsymbol{x}))^2$.

**Assumption 4.3.** For $m = 1, 2$, there exist $a_m > 0$ and $s_m \in (0, 1)$ such that $\hat{\lambda}_j^{(m)} \le a_m j^{-1/s_m}$.

Assumption 4.2 is used in [20] and is not overly restrictive as it holds for many regularization algorithms and convex, uniformly bounded function classes. In the analysis of kernel methods, Assumption 4.3 is standard [21], and is known to be equivalent to the classical covering or entropy number assumption [22]. The inverse decay rate $s_m$ measures the complexity of the RKHS, with larger values corresponding to more complex function spaces.

**Theorem 4.4.** *Let $\hat{h}$ be any element of $\mathcal{H}$ satisfying $P_n(y - \hat{h}(\boldsymbol{x}))^2 = \inf_{h \in \mathcal{H}} P_n(y - h(\boldsymbol{x}))^2$. Under Assumptions 4.2 and 4.3, for any $\eta > 0$, with probability at least $1 - 5e^{-\eta}$,*

$$P(y - \hat{h}(\boldsymbol{x}))^2 - P(y - h^*(\boldsymbol{x}))^2 = O\left(n^{-\frac{1}{1+\max\{s_1, s_2\}}}\right).$$

Theorem 4.4 suggests that the convergence rate of the excess risk depends on the decay rates of the eigenvalues of two Gram matrices $K^{(1)}$ and $K^{(2)}$. The inverse decay rate $s_1$ of the eigenvalues of $K^{(1)} = \frac{1}{n}(k_1(\boldsymbol{f_s}(\boldsymbol{x}_i), \boldsymbol{f_s}(\boldsymbol{x}_j)))_{i,j=1,...,n}$ represents the learning efficiency using only the source features, while $s_2$ is the inverse decay rate of the eigenvalues of the Hadamard product of $K_2 = \frac{1}{n}(k_2(\boldsymbol{f_s}(\boldsymbol{x}_i), \boldsymbol{f_s}(\boldsymbol{x}_j)))_{i,j=1,...,n}$ and $K_3 = \frac{1}{n}(k_3(\boldsymbol{x}_i, \boldsymbol{x}_j))_{i,j=1,...,n}$, which addresses the effect of combining the source features and original input. While rigorous discussion on the relationship between the spectra of two Gram matrices $K_2, K_3$ and their Hadamard product $K_2 \circ K_3$ seems difficult, intuitively, the smaller the overlap between the space spanned by the source features and by the original input, the smaller the overlap between $\mathcal{H}_2$ and $\mathcal{H}_3$. In other words, as the source features and original input have different information, the tensor product $\mathcal{H}_2 \otimes \mathcal{H}_3$ will be more complex, and the decay rate $1/s_2$ is expected to be larger. In Section B.1, we experimentally confirm this speculation.

# 5 Experimental Results

We demonstrate the potential of the affine model transfer through two case studies: (i) the prediction of feed-forward torque at seven joints of the robot arm [23], and (ii) the prediction of review scores and decisions of scientific papers [24]. The experimental details are presented in Section C. Additionally, two case studies in materials science are presented in Section B. The Python code is available at https://github.com/mshunya/AffineTL.

## 5.1 Kinematics of the Robot Arm

We experimentally investigated the learning performance of the affine model transfer, compared to several existing methods. The objective of the task is to predict the feed-forward torques, required to follow the desired trajectory, at seven different joints of the SARCOS robot arm [23]. Twenty-one features representing the joint position, velocity, and acceleration were used as the input $\boldsymbol{x}$. The target task is to predict the torque value at one joint. The representations encoded in the intermediate layer of the source neural network for predicting the other six joints were used as the source features $\boldsymbol{f_s} \in \mathbb{R}^{16}$. The experiments were conducted with seven different tasks (denoted as Torque 1-7) corresponding to the seven joints. For each target task, a training set of size $n \in \{5, 10, 15, 20, 30, 40, 50\}$ was randomly constructed 20 times, and the performances were evaluated using the test data.

Table 1: Performance on predicting the torque values at the first and seventh joints of the SARCOS robot arm. The mean and standard deviation of the RMSE are reported for varying numbers of training samples. For each task and $n$, the case with the smallest mean RMSE is indicated by the bold type. An asterisk indicates a case where the RMSEs of 20 independent experiments were significantly improved over **Direct** at the 1% significance level, according to the Welch's t-test. $d$ represent the dimension of the original input $x$ (i.e., $d = 21$).

| Target | Model | Number of training samples | | | | | | |
| | | $n < d$ | | | $n \approx d$ | $n > d$ | | |
| | | 5 | 10 | 15 | 20 | 30 | 40 | 50 |
| Torque 1 | Direct | $21.3 \pm 2.04$ | $18.9 \pm 2.11$ | $\mathbf{17.4 \pm 1.79}$ | $15.8 \pm 1.70$ | $13.7 \pm 1.26$ | $12.2 \pm 1.61$ | $10.8 \pm 1.23$ |
| | Only source | $24.0 \pm 6.37$ | $22.3 \pm 3.10$ | $21.0 \pm 2.49$ | $19.7 \pm 1.34$ | $18.5 \pm 1.92$ | $17.6 \pm 1.59$ | $17.3 \pm 1.31$ |
| | Augmented | $21.8 \pm 2.88$ | $19.2 \pm 1.37$ | $17.8 \pm 2.30$ | $\mathbf{15.7 \pm 1.53}$ | $\mathbf{13.3 \pm 1.19}$ | $\mathbf{11.9 \pm 1.37}$ | $\mathbf{10.7 \pm 0.954}$ |
| | HTL-offset | $23.7 \pm 6.50$ | $21.2 \pm 3.85$ | $19.8 \pm 3.23$ | $17.8 \pm 2.35$ | $16.2 \pm 3.31$ | $15.0 \pm 3.16$ | $15.1 \pm 2.76$ |
| | HTL-scale | $23.3 \pm 4.47$ | $22.1 \pm 5.31$ | $20.4 \pm 3.84$ | $18.5 \pm 2.72$ | $17.6 \pm 2.41$ | $16.9 \pm 2.10$ | $16.7 \pm 1.74$ |
| | AffineTL-full | $\mathbf{21.2 \pm 2.23}$ | $\mathbf{18.8 \pm 1.31}$ | $18.6 \pm 2.83$ | $15.9 \pm 1.65$ | $13.7 \pm 1.53$ | $12.3 \pm 1.45$ | $11.1 \pm 1.12$ |
| | AffineTL-const | $21.2 \pm 2.21$ | $18.8 \pm 1.44$ | $17.7 \pm 2.44$ | $15.9 \pm 1.58$ | $13.4 \pm 1.15$ | $12.2 \pm 1.54$ | $10.9 \pm 1.02$ |
| | Fine-tune | $25.0 \pm 7.11$ | $20.5 \pm 3.33$ | $18.6 \pm 2.10$ | $17.6 \pm 2.55$ | $14.1 \pm 1.39$ | $12.6 \pm 1.13$ | $11.1 \pm 1.03$ |
| | MAML | $29.8 \pm 12.3$ | $22.5 \pm 3.21$ | $20.8 \pm 2.12$ | $20.3 \pm 3.14$ | $16.7 \pm 3.00$ | $14.4 \pm 1.85$ | $13.4 \pm 1.19$ |
| | $L^2$-SP | $24.9 \pm 7.09$ | $20.5 \pm 3.30$ | $18.8 \pm 2.04$ | $18.0 \pm 2.45$ | $14.5 \pm 1.36$ | $13.0 \pm 1.13$ | $11.6 \pm 0.983$ |
| | PAC-Net | $25.2 \pm 8.68$ | $22.7 \pm 5.60$ | $20.7 \pm 2.65$ | $20.1 \pm 2.16$ | $18.5 \pm 2.77$ | $17.6 \pm 1.85$ | $17.1 \pm 1.38$ |
| Torque 7 | Direct | $2.66 \pm 0.307$ | $2.13 \pm 0.420$ | $1.85 \pm 0.418$ | $1.54 \pm 0.353$ | $1.32 \pm 0.200$ | $1.18 \pm 0.138$ | $1.05 \pm 0.111$ |
| | Only source | $2.31 \pm 0.618$ | $*1.73 \pm 0.560$ | $*1.49 \pm 0.513$ | $*1.22 \pm 0.269$ | $*1.09 \pm 0.232$ | $*0.969 \pm 0.144$ | $*0.927 \pm 0.170$ |
| | Augmented | $2.47 \pm 0.406$ | $1.90 \pm 0.515$ | $1.67 \pm 0.552$ | $*1.31 \pm 0.214$ | $1.16 \pm 0.225$ | $*0.984 \pm 0.149$ | $*0.897 \pm 0.138$ |
| | HTL-offset | $2.29 \pm 0.621$ | $*1.69 \pm 0.507$ | $*1.49 \pm 0.513$ | $*1.22 \pm 0.269$ | $*1.09 \pm 0.233$ | $*0.969 \pm 0.144$ | $*0.925 \pm 0.171$ |
| | HTL-scale | $2.32 \pm 0.599$ | $*1.71 \pm 0.516$ | $1.51 \pm 0.513$ | $*1.24 \pm 0.271$ | $*1.12 \pm 0.234$ | $*0.999 \pm 0.175$ | $0.948 \pm 0.172$ |
| | AffineTL-full | $*2.23 \pm 0.554$ | $*1.71 \pm 0.501$ | $\mathbf{*1.45 \pm 0.458}$ | $*1.21 \pm 0.256$ | $*1.06 \pm 0.219$ | $*0.974 \pm 0.164$ | $\mathbf{*0.870 \pm 0.121}$ |
| | AffineTL-const | $*2.30 \pm 0.565$ | $*1.73 \pm 0.420$ | $*1.48 \pm 0.527$ | $\mathbf{*1.20 \pm 0.243}$ | $\mathbf{*1.04 \pm 0.217}$ | $\mathbf{*0.963 \pm 0.161}$ | $*0.884 \pm 0.136$ |
| | Fine-tune | $*2.33 \pm 0.511$ | $*1.62 \pm 0.347$ | $*1.35 \pm 0.340$ | $*1.12 \pm 0.165$ | $*0.959 \pm 0.12$ | $*0.848 \pm 0.0824$ | $*0.790 \pm 0.0547$ |
| | MAML | $2.54 \pm 1.29$ | $1.90 \pm 0.507$ | $1.67 \pm 0.313$ | $1.63 \pm 0.282$ | $1.28 \pm 0.272$ | $1.20 \pm 0.199$ | $1.06 \pm 0.111$ |
| | $L^2$-SP | $*2.33 \pm 0.509$ | $*1.65 \pm 0.378$ | $*1.35 \pm 0.340$ | $*1.12 \pm 0.165$ | $*0.968 \pm 0.114$ | $*0.858 \pm 0.0818$ | $*0.802 \pm 0.0535$ |
| | PAC-Net | $2.24 \pm 0.706$ | $*1.61 \pm 0.394$ | $*1.43 \pm 0.389$ | $*1.24 \pm 0.177$ | $*1.18 \pm 0.100$ | $1.13 \pm 0.0726$ | $1.100 \pm 0.0589$ |

The following seven methods were compared, including two existing HTL procedures:

**Direct**   Train a model using the target input $x$ with no transfer.

**Only source**   Train a model $g(f_s)$ using only the source feature $f_s$.

**Augmented**   Perform a regression with the augmented input vector concatenating $x$ and $f_s$.

**HTL-offset** [10]   Calculate the transformed output $z_i = y_i - g_{\text{only}}(f_s)$ where $g_{\text{only}}(f_s)$ is the model pre-trained using **Only source**, and train an additional model with input $x_i$ to predict $z_i$.

**HTL-scale** [12]   Calculate the transformed output $z_i = y_i/g_{\text{only}}(f_s)$, and train an additional model with input $x_i$ to predict $z_i$.

**AffineTL-full**   Train the model $g_1 + g_2 \cdot g_3$.

**AffineTL-const**   Train the model $g_1 + g_3$.

Kernel ridge regression with the Gaussian kernel $\exp(-\|x - x'\|^2/2\ell^2)$ was used for each procedure. The scale parameter $\ell$ was fixed to the square root of the dimension of the input. The regularization parameter in the kernel ridge regression and $\lambda_\alpha, \lambda_\beta,$ and $\lambda_\gamma$ in the affine model transfer were selected through 5-fold cross-validation. In addition to the seven feature-based methods, four weight-based TL methods were evaluated: fine-tuning, MAML [25], $L^2$-SP [26], and PAC-Net [27].

Table 1 summarizes the prediction performance of the seven different procedures for varying numbers of training samples in two representative tasks: Torque 1 and Torque 7. The joint of Torque 1 is located closest to the root of the arm. Therefore, the learning task for Torque 1 is less relevant to those for the other joints, and the transfer from Torque 2–6 to Torque 1 would not work. In fact, as shown in Table 1, no method showed a statistically significant improvement to **Direct**. In particular, **Only source** failed to acquire predictive ability, and **HTL-offset** and **HTL-scale** likewise showed poor prediction performance owing to the negative effect of the failure in the variable transformation. In contrast, the two affine transfer models showed almost the same predictive performance as **Direct**, which is expressed as its submodel, and successfully suppressed the occurrence of negative transfer.

Because Torque 7 was measured at the joint closest to the end of the arm, its value strongly depends on those at the other six joints, and the procedures with the source features were more effective than in the other tasks. In particular, **AffineTL** achieved the best performance among the other feature-based methods. This is consistent with the theoretical result that the transfer capability of the affine model transfer can be further improved when the risk of learning using only the source features is sufficiently small.

In Table S.3 in Section C.1, we present the results for all tasks. In most cases, **AffineTL** achieved the best performance among the feature-based methods. In several other cases, **Direct** produced the best results; in almost all cases, **Only source** and the two HTLs showed no advantage over **AffineTL**. Comparing the weight-based and feature-based methods, we noticed that the weight-based methods showed higher performance with large sample sizes. Nevertheless, in scenarios with extremely small sample sizes (e.g., $n = 5$ or 10), **AffineTL** exhibited comparable or even superior performance.

The strength of our method compared to weight-based TLs including fine-tuning is that it does not degrade its performance in cases where cross-domain relationships are weak. While fine-tuning outperformed our method in cases of Torque 7, the performance of fine-tuning was significantly degraded as the source-target relationship became weaker, as seen in Torque 1 case. In contrast, our method was able to avoid negative transfer even for such cases. This characteristic is particularly beneficial because, in many cases, the degree of relatedness between the domains is not known in advance. Furthermore, weight-based methods can sometimes be unsuitable, especially when transferring knowledge from large models, such as LLMs. In these scenarios, fine-tuning all parameters is unfeasible, and feature-based TL is preferred. Our approach often outperforms other feature-based methods.

## 5.2 Evaluation of Scientific Documents

Through a case study in natural language processing, we compare the performance of the affine model transfer with that of ordinary feature extraction-based TL and show the advantage of being able to estimate domain shift and domain-specific factors separately.

We used SciRepEval [24], a benchmark dataset of scientific documents. The dataset consists of abstracts, review scores, and decision statuses of papers submitted to various machine learning conferences. We focused on two primary tasks: a regression task to predict the average review score, and a binary classification task to determine the acceptance or rejection status of each paper. The original input $x$ was represented by a two-gram bag-of-words vector of the abstract. For the source features $f_s$, we utilized text embeddings of the abstract generated by the pre-trained language models; BERT [28], SciBERT [29], T5 [30], and GPT-3 [31]. In the affine model transfer, we employed neural networks with two hidden layers to model $g_1$ and $g_2$, and a linear model for $g_3$. For comparison, we also evaluated the performance of the ordinary feature extraction-based TL using a two-layer neural network with $f_s$ as inputs. We used 8,166 training samples and evaluated the performance of the model on 2,043 test samples.

Table 2 shows the root mean square error (RMSE) for the regression task and accuracy for the classification task. In the regression tasks, the RMSEs of the affine model transfer were significantly improved over the ordinary feature extraction for the four types of text feature embedding. We also observed the improvements in accuracy for the classification task even though the affine model transfer was derived on the basis of regression settings. While the pre-trained language models have the remarkable ability to represent text quality and structure, their representation ability to perform prediction tasks for machine learning documents is not sufficient. The affine model transfer effectively bridged this gap by learning the additional target-specific factor via the target task, resulting in improved prediction performance in both regression and classification tasks.

Table 3 provides a list of phrases that were estimated to have a positive or negative effect on the review scores. Because we restricted a network to output positive values for $g_2$, the influence of each phrase could be inferred from the estimated coefficients of the linear model $g_3$. Specifically, phrases such as *"tasks including"* and *"new state"* were estimated to have positive influences on the predicted score. These phrases often appear in contexts such as *"demonstrated on a wide range of tasks including"* or *"establishing a new state-of-the-art result,"*, suggesting that superior experimental results tend to yield higher peer review scores. In addition, the phrase *"theoretical analysis"* was also identified to have a positive effect on the review score, reflecting the significance of theoretical validation in machine learning research. On the contrary, general phrases with broader meanings such as *"recent advances"* and *"machine learning,"* contributed to lower scores. This observation suggests the importance of explicitly stating the novelty and uniqueness of research findings and refraining from using generic terminologies.

Table 2: Prediction performance of peer review scores and acceptance/rejection for submitted papers. The mean and standard deviation of the RMSE and accuracy are reported for the affine model transfer (AffineTL) and feature extraction (FE). Definitions of asterisks and boldface letters are the same as in Table 1.

| | | Regression | Classification |
|---|---|---|---|
| BERT | FE | $1.3086 \pm 0.0035$ | $0.6250 \pm 0.0217$ |
| | AffineTL | $*1.3069 \pm 0.0042$ | $0.6252 \pm 0.0163$ |
| SciBERT | FE | $1.2856 \pm 0.0144$ | $\mathbf{0.6520 \pm 0.0106}$ |
| | AffineTL | $*1.2797 \pm 0.0122$ | $0.6507 \pm 0.0124$ |
| T5 | FE | $1.3486 \pm 0.0175$ | $0.6344 \pm 0.0079$ |
| | AffineTL | $*1.3442 \pm 0.0030$ | $\mathbf{0.6366 \pm 0.0065}$ |
| GPT-3 | FE | $1.3284 \pm 0.0138$ | $0.6279 \pm 0.0181$ |
| | AffineTL | $*\mathbf{1.3234 \pm 0.0140}$ | $*\mathbf{0.6386 \pm 0.0095}$ |

Table 3: Phrases with the top and bottom ten regression coefficients for $g_3$ in the affine transfer model for the regression task with SciBERT.

| | Positive | Negative |
|---|---|---|
| 1 | tasks including | recent advances |
| 2 | new state | novel approach |
| 3 | high quality | latent space |
| 4 | recently proposed | learning approach |
| 5 | latent variable | neural architecture |
| 6 | number parameters | machine learning |
| 7 | theoretical analysis | attention mechanism |
| 8 | policy gradient | reinforcement learning |
| 9 | inductive bias | proposed framework |
| 10 | image generation | descent sgd |

As illustrated in this example, integrating modern deep learning techniques and highly interpretable transfer models through the mechanism of the affine model transfer not only enhances prediction performance, but also provides valuable insights into domain-specific factors.

### 5.3 Case Studies in Materials Science

We conducted two additional case studies, both of which pertain to scientific tasks in the field of materials science. One experiment aims to examine the relationship between qualitative differences in source features and learning behavior of the affine model transfer. In the other experiment, we demonstrate the potential utility of the affine model transfer as a calibration tool bridging computational models and real-world systems. In particular, we highlight the benefits of separately modeling and estimating domain-specific factors through a case study in polymer chemistry. The objective is to predict the specific heat capacity at constant pressure of any given organic polymer with its chemical structure in the polymer's repeating unit. Specifically, we conduct TL to bridge the gap between experimental values and physical properties calculated from molecular dynamics simulations. The details are shown in Section B in Supplementary Material,

## 6 Conclusions

In this study, we introduced a general class of TL based on affine model transformations, and clarified their learning capability and applicability. The proposed affine model transformation was shown to be an optimal class that minimizes the expected squared loss in the HTL procedure. The model is contrasted with widely applied TL methods, such as re-using features from pre-trained models, which lack theoretical foundation. The affine model transfer is model-agnostic; it is easily combined with any machine learning models, features, and physical models. Furthermore, in the model, domain-specific factors are involved in incorporating the source features. From this property, the affine transfer has the ability to handle domain common and unique factors simultaneously and separately.

The advantages of the model were verified theoretically and experimentally in this study. We showed theoretical results on the generalization bound and excess risk bound when the regression tasks are solved by kernel methods. It is shown that if the source feature is strongly related to the target domain, the convergence rate of the generalization bound is improved from naive learning. The excess risk of the proposed TL is evaluated using the eigen-decay of the product kernel, which also illustrates the effect of the overlap between the source and target tasks. In our numerical studies, the affine model transfer generally outperforms in test errors when the target and source tasks have a similarity. We have also seen in the example of NLP that the proposed affine model transfer can identify the (non-)valuable phrases for high-quality papers. This can be done by the affine representation of cross-domain shift and domain-specific factors in our model.

## Acknowledgments and Disclosure of Funding

This work was supported by JST SPRING Grant No. JPMJSP2104, JST CREST Grants No. JP-MJCR22O3 and No. JPMJCR19I3, MEXT KAKENHI Grant-in-Aid for Scientific Research on Innovative Areas (Grant No. 19H50820), the Grant-in-Aid for Scientific Research (A) (Grant No. 19H01132) and Grant-in-Aid for Research Activity Start-up (Grant No. 23K19980) from the Japan Society for the Promotion of Science (JSPS), and the MEXT Program for Promoting Researches on the Supercomputer Fugaku (No. hp210264).

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

# Supplementary Material
# Transfer Learning with Affine Model Transformation

## A  Other Perspectives on Affine Model Transfer

### A.1  Transformation Functions for General Loss Functions

Here we discuss the optimal transformation function for general loss functions.

Let $\ell(y, y') \geq 0$ be a convex loss function that returns zero if and only if $y = y'$, and let $g^*(\boldsymbol{x})$ be the optimal predictor that minimizes the expectation of $\ell$ with respect to the distribution $p_t$ followed by $\boldsymbol{x}$ and $y$ transformed by $\phi$:

$$g^*(\boldsymbol{x}) = \arg\min_g \mathbb{E}_{p_t}\big[\ell(g(\boldsymbol{x}), \phi_{f_s}(y))\big].$$

The function $g$ that minimizes the expected loss

$$\mathbb{E}_{p_t}\big[\ell(g(\boldsymbol{x}), \phi_{f_s}(y))\big] = \iint \ell(g(\boldsymbol{x}), \phi_{f_s}(y))p_t(\boldsymbol{x}, y)\mathrm{d}\boldsymbol{x}\mathrm{d}y$$

should be a solution to the Euler-Lagrange equation

$$\frac{\partial}{\partial g(\boldsymbol{x})} \int \ell(g(\boldsymbol{x}), \phi_{f_s}(y))p_t(\boldsymbol{x}, y)\mathrm{d}y = \int \frac{\partial}{\partial g(\boldsymbol{x})}\ell(g(\boldsymbol{x}), \phi_{f_s}(y))p_t(y|\boldsymbol{x})\mathrm{d}y\, p_t(\boldsymbol{x}) = 0. \quad \text{(S.1)}$$

Denote the solution of Eq. (S.1) by $G(\boldsymbol{x}; \phi_{f_s})$. While $G$ depends on the loss $\ell$ and distribution $p_t$, we omit those from the argument for notational simplicity. Using this function, the minimizer of the expected loss $\mathbb{E}_{\boldsymbol{x}, y}[\ell(g(\boldsymbol{x}), y)]$ can be expressed as $G(\boldsymbol{x}; \mathrm{id})$, where $\mathrm{id}$ represents the identity function.

Here, we consider the following assumption to hold, which generalizes Assumption 2.3 in the main text:

**Assumption 2.3(b).** *For any distribution on the target domain $p_t(\boldsymbol{x}, y)$ and all $\boldsymbol{x} \in \mathcal{X}$, the following relationship holds:*

$$\psi_{f_s}(g^*(\boldsymbol{x})) = \arg\min_g \mathbb{E}_{\boldsymbol{x}, y}[\ell(g(\boldsymbol{x}), y)].$$

*Equivalently, the transformation functions $\phi_{f_s}$ and $\psi_{f_s}$ satisfy*

$$\psi_{f_s}\big(G(\boldsymbol{x}; \phi_{f_s})\big) = G(\boldsymbol{x}; \mathrm{id}). \quad \text{(S.2)}$$

Assumption 2.3(b) states that if the optimal predictor $G(\boldsymbol{x}; \phi_{f_s})$ for the data transformed by $\phi$ is given to the model transformation function $\psi$, it is consistent with the overall optimal predictor $G(\boldsymbol{x}; \mathrm{id})$ in the target region in terms of the loss function $\ell$. We consider all pairs of $\psi$ and $\phi$ that satisfy this consistency condition.

Here, let us consider the following proposition:

**Proposition A.1.** *Under Assumption 2.1, 2.2 and 2.3(b), $\psi_{f_s}^{-1} = \phi_{f_s}$.*

*Proof.* The proof is analogous to that of Theorem 2.4 in Section D.1. For any $y_0 \in \mathcal{Y}$, let $p_t(y|\boldsymbol{x}) = \delta_{y_0}$. Combining this with Eq. (S.1) leads to

$$\frac{\partial}{\partial g(\boldsymbol{x})}\ell(g(\boldsymbol{x}), \phi_{f_s}(y_0)) = 0 \ (\forall y_0 \in \mathcal{Y}).$$

Because $\ell(y, y')$ returns the minimum value zero if and only if $y = y'$, we obtain $G(\boldsymbol{x}; \phi_{f_s}) = \phi_{f_s}(y_0)$. Similarly, we have $G(\boldsymbol{x}; \mathrm{id}) = y_0$. From these two facts and Assumption 2.3(b), we have $\psi_{f_s}(\phi_{f_s}(y_0)) = y_0$, proving that the proposition is true. $\qquad\square$

Proposition A.1 indicates that the first statement of Theorem 2.4 holds for general loss functions. However, the second claim of Theorem 2.4 generally depends on the type of loss function. Through the following examples, we describe the optimal class of transformation functions for several loss functions.

**Example 1** (Squared loss). Let $\ell(y, y') = |y - y'|^2$. As a solution of Eq. (S.1), we can see that the optimal predictor is the conditional expectation $\mathbb{E}_{p_t}[\phi_{f_s}(Y)|\boldsymbol{X} = \boldsymbol{x}]$. As discussed in Section 2.1, the transformation functions $\phi_{f_s}$ and $\psi_{f_s}$ should be affine transformations.

**Example 2** (Absolute loss). Let $\ell(y, y') = |y - y'|$. Substituting this into Eq. (S.1), we have

$$0 = \int \frac{\partial}{\partial g(\boldsymbol{x})} |g(\boldsymbol{x}) - \phi_{f_s}(y)| p_t(y|\boldsymbol{x}) \mathrm{d}y$$

$$= \int \mathrm{sign}\big(g(\boldsymbol{x}) - \phi_{f_s}(y)\big) p_t(y|\boldsymbol{x}) \mathrm{d}y$$

$$= \int_{\phi_{f_s}(y) \geq g(\boldsymbol{x})} p_t(y|\boldsymbol{x}) \mathrm{d}y - \int_{\phi_{f_s}(y) < g(\boldsymbol{x})} p_t(y|\boldsymbol{x}) \mathrm{d}y.$$

Assuming that $\phi_{f_s}$ is monotonically increasing, we have

$$0 = \int_{y \geq \phi_{f_s}^{-1}(g(\boldsymbol{x}))} p_t(y|\boldsymbol{x}) \mathrm{d}y - \int_{y < \phi_{f_s}^{-1}(g(\boldsymbol{x}))} p_t(y|\boldsymbol{x}) \mathrm{d}y.$$

This yields

$$\int_{\phi_{f_s}^{-1}(g(\boldsymbol{x}))}^{\infty} p_t(y|\boldsymbol{x}) \mathrm{d}y = \int_{-\infty}^{\phi_{f_s}^{-1}(g(\boldsymbol{x}))} p_t(y|\boldsymbol{x}) \mathrm{d}y.$$

The same result is obtained even if $\phi_{f_s}$ is monotonically decreasing. Consequently,

$$\phi_{f_s}^{-1}(g(\boldsymbol{x})) = \mathrm{Median}[Y|\boldsymbol{X} = \boldsymbol{x}],$$

which results in

$$G(\boldsymbol{x}; \phi_{f_s}) = \phi_{f_s}\big(\mathrm{Median}[Y|\boldsymbol{X} = \boldsymbol{x}]\big).$$

This implies that Eq. (S.2) holds for any $\phi_{f_s}$ including an affine transformation, and the function form cannot be identified. from this analysis.

**Example 3** (Exponential-squared loss). As an example where the affine transformation is not optimal, consider the loss function $\ell(y, y') = |e^y - e^{y'}|^2$. Substituting this into Eq. (S.1), we have

$$0 = \int \frac{\partial}{\partial g(\boldsymbol{x})} \big|\exp(g(\boldsymbol{x})) - \exp(\phi_{f_s}(y))\big|^2 p_t(y|\boldsymbol{x}) \mathrm{d}y$$

$$= 2\exp(g(\boldsymbol{x})) \int \big(\exp(g(\boldsymbol{x})) - \exp(\phi_{f_s}(y))\big) p_t(y|\boldsymbol{x}) \mathrm{d}y.$$

Therefore,

$$G(\boldsymbol{x}; \phi_{f_s}) = \log \mathbb{E}\big[\exp(\phi_{f_s}(Y))|\boldsymbol{X} = \boldsymbol{x}\big].$$

Consequently, Eq. (S.2) becomes

$$\log \mathbb{E}\big[\exp(\phi_{f_s}(Y))\big] = \phi_{f_s}\big(\log \mathbb{E}\big[\exp(Y)\big]\big).$$

Even if $\phi_{f_s}$ is an affine transformation, this equation does not generally hold.

## A.2 Analysis of the Optimal Function Class Based on the Upper Bound of the Estimation Error

Here, we discuss the optimal class for the transformation function based on the upper bound of the estimation error.

In addition to Assumptions 2.1 and 2.2, we assume the following in place of Assumption 2.3:

**Assumption A.2.** The transformation functions $\phi$ and $\psi$ are Lipschitz continuous with respect to the first argument, i.e., there exist constants $\mu_\phi$ and $\mu_\psi$ such that,

$$\phi(a, c) - \phi(a', c) \leq \mu_\phi \|a - a'\|_2, \quad \psi(b, c) - \psi(b', c) \leq \mu_\psi \|b - b'\|_2,$$

for any $a, a' \in \mathcal{Y}$ and $b, b' \in \mathbb{R}$ with any given $c \in \mathcal{F}_s$.

Note that each Lipschitz constant is a function of the second argument $\boldsymbol{f_s}$, i.e., $\mu_\phi = \mu_\phi(\boldsymbol{f_s})$ and $\mu_\psi = \mu_\psi(\boldsymbol{f_s})$.

Under Assumptions 2.1, 2.2 and A.2, the estimation error is upper bounded as follows:

$$
\begin{aligned}
\mathop{\mathbb{E}}_{\boldsymbol{x},y}\left[\,|f_t(\boldsymbol{x}) - \hat{f}_t(\boldsymbol{x})|^2\right] &= \mathop{\mathbb{E}}_{\boldsymbol{x},y}\left[\,\big|\psi(g(\boldsymbol{x}), \boldsymbol{f_s}(\boldsymbol{x})) - \psi(\hat{g}(\boldsymbol{x}), \boldsymbol{f_s}(\boldsymbol{x}))\big|^2\right] \\
&\leq \mathop{\mathbb{E}}_{\boldsymbol{x},y}\left[\mu_\psi(\boldsymbol{f_s}(\boldsymbol{x}))^2 \big|g(\boldsymbol{x}) - \hat{g}(\boldsymbol{x})\big|^2\right] \\
&\leq 3\mathop{\mathbb{E}}_{\boldsymbol{x},y}\Big[\mu_\psi(\boldsymbol{f_s}(\boldsymbol{x}))^2\big(\big|g(\boldsymbol{x}) - \phi(f_t(\boldsymbol{x}), f_s(\boldsymbol{x}))\big|^2 \\
&\qquad\qquad\qquad\qquad + \big|\phi(f_t(\boldsymbol{x}), \boldsymbol{f_s}(\boldsymbol{x})) - \phi(y, \boldsymbol{f_s}(\boldsymbol{x}))\big|^2 \\
&\qquad\qquad\qquad\qquad + \big|\phi(y, \boldsymbol{f_s}(\boldsymbol{x})) - \hat{g}(\boldsymbol{x})\big|^2\big)\Big] \\
&\leq 3\mathop{\mathbb{E}}_{\boldsymbol{x},y}\left[\mu_\psi(\boldsymbol{f_s}(\boldsymbol{x}))^2\big|\psi^{-1}(f_t(\boldsymbol{x}), \boldsymbol{f_s}(\boldsymbol{x})) - \phi(f_t(x), \boldsymbol{f_s}(\boldsymbol{x}))\big|^2\right] \\
&\qquad + 3\mathop{\mathbb{E}}_{\boldsymbol{x},y}\left[\mu_\psi(\boldsymbol{f_s}(\boldsymbol{x}))^2\mu_\phi(\boldsymbol{f_s}(\boldsymbol{x}))^2\big|f_t(\boldsymbol{x}) - y\big|^2\right] \\
&\qquad + 3\mathop{\mathbb{E}}_{\boldsymbol{x},y}\left[\mu_\psi(\boldsymbol{f_s}(\boldsymbol{x}))^2\big|z - \hat{g}(\boldsymbol{x})\big|^2\right] \\
&= 3\mathop{\mathbb{E}}_{\boldsymbol{x},y}\left[\mu_\psi(\boldsymbol{f_s}(\boldsymbol{x}))^2\big|\psi^{-1}(f_t(\boldsymbol{x}), \boldsymbol{f_s}(\boldsymbol{x})) - \phi(f_t(\boldsymbol{x}), \boldsymbol{f_s}(\boldsymbol{x}))\big|^2\right] \\
&\qquad + 3\sigma^2\mathop{\mathbb{E}}_{\boldsymbol{x},y}\left[\mu_\psi(\boldsymbol{f_s}(\boldsymbol{x}))^2\mu_\phi(\boldsymbol{f_s}(\boldsymbol{x}))^2\right] \\
&\qquad + 3\mathop{\mathbb{E}}_{\boldsymbol{x},y}\left[\mu_\psi(\boldsymbol{f_s}(\boldsymbol{x}))^2\big|z - \hat{g}(\boldsymbol{x})\big|^2\right].
\end{aligned}
$$

The derivation of this inequality is based on [12]. We use the Lipschitz property of $\psi$ and $\phi$ for the first and third inequalities, and the second inequality comes from the inequality $(a - d)^2 \leq 3(a - b)^2 + 3(b - c)^2 + 3(c - d)^2$ for $a, b, c, d \in \mathbb{R}$.

According to this inequality, the upper bound of the estimation error is decomposed into three terms: the discrepancy between the two transformation functions, the variance of the noise, and the estimation error for the transformed data. Although it is intractable to find the optimal solution of $\phi, \psi, \hat{g}$ that minimizes all these terms together, it is possible to find a solution that minimizes the first and second terms expressed as the functions of $\phi$ and $\psi$ only. Obviously, the first term, which represents the discrepancy between the two transformation functions, reaches its minimum (zero) when $\phi_{f_s} = \psi_{f_s}^{-1}$. The second term, which is related to the variance of the noise, is minimized when the differential coefficient $\frac{\partial}{\partial u}\psi_{f_s}(u)$ is a constant, i.e., when $\psi_{f_s}$ is a linear function. This is verified as follows. From $\phi_{f_s} = \psi_{f_s}^{-1}$ and the continuity of $\psi_{f_s}$, it follows that

$$
\mu_\psi = \max\left|\frac{\partial}{\partial u}\psi_{f_s}(u)\right|, \quad \mu_\phi = \max\left|\frac{\partial}{\partial u}\psi_{f_s}^{-1}(u)\right| = \frac{1}{\min\left|\frac{\partial}{\partial u}\psi_{f_s}(u)\right|},
$$

and thus the product $\mu_\phi\mu_\psi$ takes the minimum value (one) when the maximum and minimum of the differential coefficient are the same. Therefore, we can write

$$
\phi(y, \boldsymbol{f_s}) = \frac{y - g_1(\boldsymbol{f_s})}{g_2(\boldsymbol{f_s})}, \quad \psi(g(x), \boldsymbol{f_s}) = g_1(\boldsymbol{f_s}) + g_2(\boldsymbol{f_s})g(\boldsymbol{x}),
$$

where $g_1, g_2 : \mathcal{F}_s \to \mathbb{R}$ are arbitrarily functions. Thus, the minimization of the third term in the upper bound of the estimation error can be expressed as

$$
\min_{g_1, g_2, g} \mathop{\mathbb{E}}_{\boldsymbol{x},y} |y - g_1(\boldsymbol{f_s}) - g_2(\boldsymbol{f_s})g(\boldsymbol{x})|^2.
$$

As a result, the suboptimal function class for the upper bound of the estimated function is given as

$$
\mathcal{H} = \big\{x \mapsto g_1(\boldsymbol{f_s}) + g_2(\boldsymbol{f_s}) \cdot g_3(\boldsymbol{x}) \mid g_1 \in \mathcal{G}_1, g_2 \in \mathcal{G}_2, g_3 \in \mathcal{G}_3\big\}.
$$

This is the same function class derived in Section 2.1.

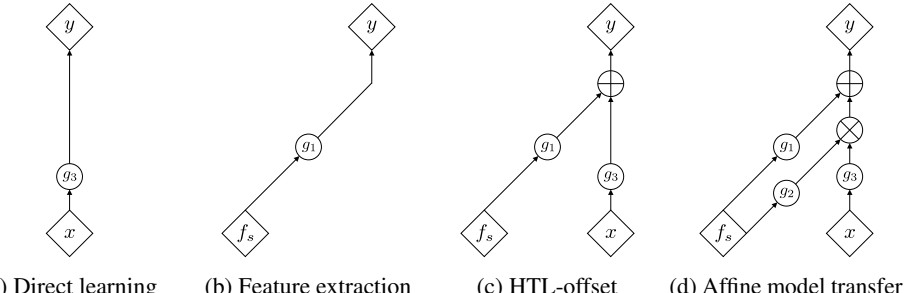

| (a) Direct learning | (b) Feature extraction | (c) HTL-offset | (d) Affine model transfer |

Figure S.1: Model architectures for the affine model transfer and related procedures. (a) Direct learning predicts outputs using only the original inputs $x$, while (b) feature extraction-based neural transfer predicts outputs using only the source features $\boldsymbol{f_s}$. (c) The HTL procedure proposed in [10] (HTL-offset) constructs the predictor as the sum of $g_1(\boldsymbol{f_s})$ and $g_3(\boldsymbol{x})$. (d) The affine model transfer encompasses these procedures, computing $g_1$ and $g_2$ as functions of the source features and constructing the predictor as an affine combination with $g_3$.

## B  Additional Experiments

### B.1  Eigenvalue Decay of the Hadamard Product of Two Gram Matrices

We experimentally investigated how the decay rate $s_2$ in Theorem 4.4 is related to the overlap degree in the spaces spanned by the original input $\boldsymbol{x}$ and source features $\boldsymbol{f_s}$.

For the original input $\boldsymbol{x} \in \mathbb{R}^{100}$, we randomly constructed a set of 10 orthonormal bases, and then generated 100 samples from their spanning space. For the source features $\boldsymbol{f_s} \in \mathbb{R}^{100}$, we selected $d$ bases randomly from the 10 orthonormal bases selected for $\boldsymbol{x}$ and the remaining $10 - d$ bases from their orthogonal complement space. We then generated 100 samples of $\boldsymbol{f_s}$ from the space spanned by these 10 bases. The overlap number $d$ can be regarded as the degree of overlap of two spaces spanned by the samples of $\boldsymbol{x}$ and $\boldsymbol{f_s}$. We generated the 100 different sample sets of $\boldsymbol{x}$ and $\boldsymbol{f_s}$.

We calculated the Hadamard product of the Gram matrices $K_2$ and $K_3$ using the samples of $\boldsymbol{x}$ and $\boldsymbol{f_s}$, respectively. For the computation of $K_2$ and $K_3$, all combinations of the following five kernels were tested:

**Linear kernel** $\ k(\boldsymbol{x}, \boldsymbol{x}') = \dfrac{\boldsymbol{x}^\top \boldsymbol{x}}{2\gamma^2} + 1,$

**Matérn kernel** $\ k(\boldsymbol{x}, \boldsymbol{x}') = \dfrac{2^{1-\nu}}{\Gamma(\nu)} \left( \dfrac{\sqrt{2\nu}\|\boldsymbol{x} - \boldsymbol{x}'\|_2}{\gamma} \right)^\nu K_\nu \left( \dfrac{\sqrt{2\nu}\|\boldsymbol{x} - \boldsymbol{x}'\|_2}{\gamma} \right)$ for $\nu = \dfrac{1}{2}, \dfrac{3}{2}, \dfrac{5}{2}, \infty,$

where $K_\nu(\cdot)$ is a modified Bessel function and $\Gamma(\cdot)$ is the gamma function. Note that for $\nu = \infty$, the Matérn kernel is equivalent to the Gaussian RBF kernel. The scale parameter $\gamma$ of both kernels was set to $\gamma = \sqrt{\dim(\boldsymbol{x})} = \sqrt{10}$. For a given matrix $K$, the decay rate of the eigenvalues was estimated as the smallest value of $s$ that satisfies $\lambda_i \leq \|K\|_F^2 \cdot i^{-\frac{1}{s}}$ where $\|\cdot\|_F$ denotes the Frobenius norm. Note that this inequality holds for any matrices $K$ with $s = 1$ [32].

Figure S.2 shows the change of the decay rates with respect to varying $d$ for various combinations of the kernels. In all cases, the decay rate of $K_2 \circ K_3$ showed a clear trend of monotonically decreasing as the degree of overlap $d$ increases. In other words, the greater the overlap between the spaces spanned by $\boldsymbol{x}$ and $\boldsymbol{f_s}$, the smaller the decay rate, and the smaller the complexity of the RKHS $\mathcal{H}_2 \otimes \mathcal{H}_3$.

### B.2  Lattice Thermal Conductivity of Inorganic Crystals

Here, we describe the relationship between the qualitative differences in source features and the learning behavior of the affine model transfer, in contrast to ordinary feature extraction using neural networks.

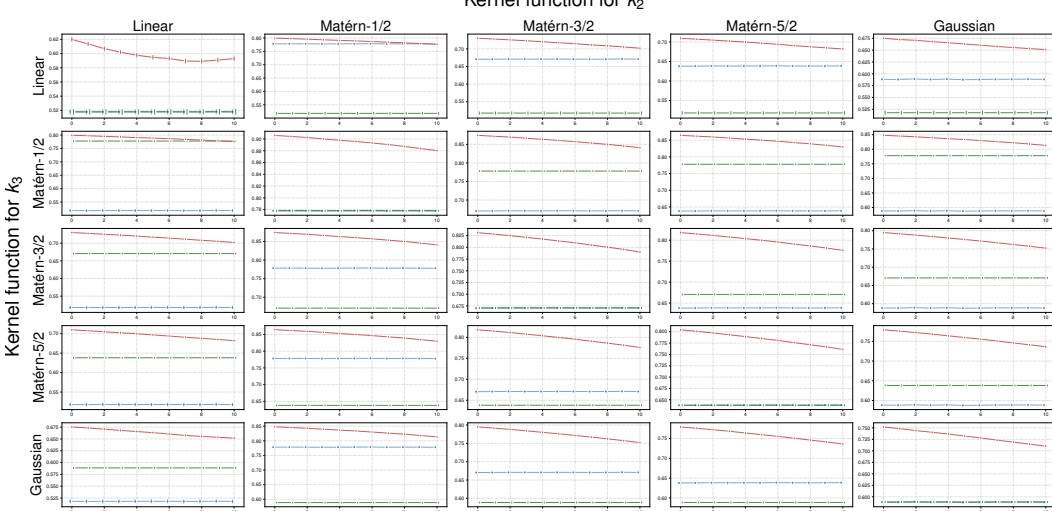

Figure S.2: Decay rates of eigenvalues of $K_2$ (blue lines), $K_3$ (green lines) and $K_2 \circ K_3$ (red lines) for all combinations of the five different kernels. The vertical axis represents the decay rate, and the horizontal axis represents the overlap dimension $d$ in the space where $\boldsymbol{x}$ and $\boldsymbol{f_s}$ are distributed.

The target task is to predict the lattice thermal conductivity (LTC) of inorganic crystalline materials, where the LTC is the amount of vibrational energy propagated by phonons in a crystal. In general, LTC can be calculated ab initio by performing many-body electronic structure calculations based on quantum mechanics. However, it is quite time-consuming to perform the first-principles calculations for thousands of crystals, which will be used as a training sample set to create a surrogate statistical model. Therefore, we perform TL with the source task of predicting an alternative, computationally tractable physical property called scattering phase space (SPS), which is known to be physically related to LTC.

### B.2.1 Data

We used the dataset from [8] that records SPS and LTC for 320 and 45 inorganic compounds, respectively. The input compounds were translated to 290-dimensional compositional descriptors using XenonPy [6, 33, 34, 35][1].

### B.2.2 Model Definition and Hyperparameter Search

Fully connected neural networks were used for both the source and target models, with a LeakyReLU activation function with $\alpha = 0.01$. The model training was conducted using the Adam optimizer [36]. Hyperparameters such as the width of the hidden layer, learning rate, number of epochs, and regularization parameters were adjusted with 5-fold cross-validation. For more details on the experimental conditions and procedure, refer to the provided Python code.

**Source Model** For the preliminary step, neural networks with three hidden layers that predict SPS were trained using 80% of the 320 samples. 100 models with different numbers of neurons were randomly generated and the top 10 source models that showed the highest generalization performance in the source domain were selected. The hidden layer width $L$ was randomly chosen from the range $[50, 100]$, and we trained a neural network with a structure of (input)-$L$-$L$-$L$-1. Each of the three hidden layers of the source model was used as an input to the transfer models, and we examined the difference in prediction performance for the three layers.

**Target Model** In the target task, an intermediate layer of a source model was used as the feature extractor. A model was trained using 40 randomly chosen samples of LTC, and its performance was evaluated with the remaining 5 samples. For each of the 10 source models, we performed the

---

[1]`https://github.com/yoshida-lab/XenonPy`

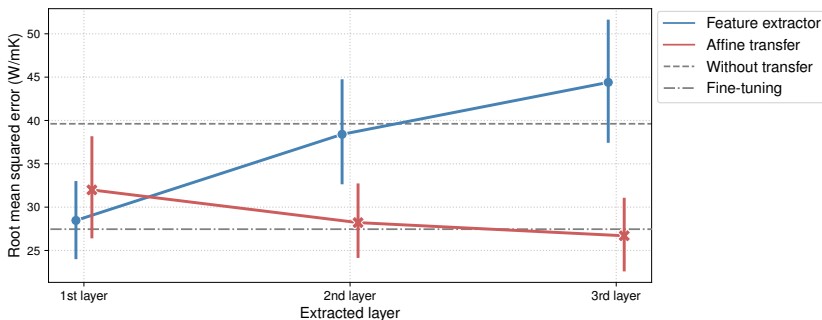

Figure S.3: Change of RMSE values between the affine transfer model and the ordinary feature extractor when using different levels of intermediate layers as the source features. The line plot shows the mean and 95% confidence interval. As a baseline, RMSE values for direct learning without transfer and fine-tuned neural networks are shown as dotted and dashed lines, respectively.

training and testing 10 times with different sample partitions and compared the mean values of RMSE among four different methods: (i) the affine model transfer using neural networks to model the three functions $g_1, g_2$ and $g_3$, (ii) a neural network using the XenonPy compositional descriptors as input without transfer, (iii) a neural network using the source features as input, and (iv) fine-tuning of the pre-trained neural networks. The width of the layers of each neural network, the number of training epochs, and the dropout rate were optimized during 5-fold cross-validation looped within each training set. For the affine model transfer, the functions $g_1$, $g_2$, and $g_3$ were modeled by neural networks. We used neural networks with one hidden layer for $g_1, g_2$ and $g_3$.

### B.2.3 Results

Figure S.3 shows the change in prediction performance of TL models using source features obtained from different intermediate layers from the first to the third layers. The affine transfer model and the ordinary feature extractor showed opposite patterns. The performance of the feature extractor improved when the first intermediate layer closest to the input layer was used as the source features and gradually degraded when layers closer to the output were used. When the third intermediate layer was used, a negative transfer occurred in the feature extractor as its performance became worse than that of the direct learning. In contrast, the affine transfer model performs better as the second and third intermediate layers closer to the output were used. The affine transfer model using the third intermediate layer reached a level of accuracy slightly better than fine-tuning, which intuitively uses more information to transfer than the extracted features.

In general, the features encoded in an intermediate layer of a neural network are more task-independent as the layer is closer to the input, and the features are more task-specific as the layer is closer to the output [9]. Because the first layer does not differ much from the original input, using both $x$ and $f_s$ in the affine model transfer does not contribute much to performance improvement. However, when using the second and third layers as the feature extractors, the use of both $x$ and $f_s$ contributes to improving the expressive power of the model, because the feature extractors have acquired different representational capabilities from the original input. In contrast, a model based only on $f_s$ from a source task-specific feature extractor could not account for data in the target domain, so its performance would become worse than direct learning without transfer, i.e., a negative transfer would occur.

### B.3 Heat Capacity of Organic Polymers

We highlight the benefits of separately modeling and estimating domain-specific factors through a case study in polymer chemistry. The objective is to predict the specific heat capacity at constant pressure $C_P$ of any given organic polymer with its chemical structure in the polymer's repeating unit. Specifically, we conduct TL to bridge the gap between experimental values and physical properties calculated from molecular dynamics (MD) simulations.

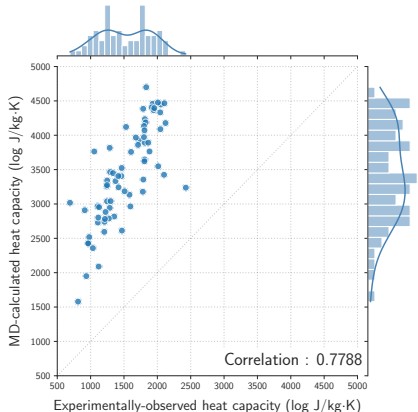

Figure S.4: MD-calculated (vertical axis) and experimental values (horizontal axis) of the specific heat capacity at constant pressure for various amorphous polymers.

Table S.1: Force field parameters that form the General AMBER force field [37] version 2 (GAFF2), and their detailed descriptions.

| Parameter | Description |
|---|---|
| mass | Atomic mass |
| $\sigma$ | Equilibrium radius of van der Waals (vdW) interactions |
| $\epsilon$ | Depth of the potential well of vdW interactions |
| charge | Atomic charge of Gasteiger model |
| $r_0$ | Equilibrium length of chemical bonds |
| $K_{\mathrm{bond}}$ | Force constant of bond stretching |
| polar | Bond polarization defined by the absolute value of charge difference between atoms in a bond |
| $\theta_0$ | Equilibrium angle of bond angles |
| $K_{\mathrm{angle}}$ | Force constant of bond bending |
| $K_{\mathrm{dih}}$ | Rotation barrier height of dihedral angles |

As shown in Figure S.4, there was a large systematic bias between experimental and calculated values; the MD-calculated properties $C_{\mathrm{P}}^{\mathrm{MD}}$ exhibited an evident overestimation with respect to their experimental values. As discussed in [38], this bias is inevitable because classical MD calculations do not reflect the presence of quantum effects in the real system. According to Einstein's theory for the specific heat in physical chemistry, the logarithmic ratio between $C_{\mathrm{P}}^{\mathrm{exp}}$ and $C_{\mathrm{P}}^{\mathrm{MD}}$ can be calibrated by the following equation:

$$\log C_{\mathrm{P}}^{\mathrm{exp}} = \log C_{\mathrm{P}}^{\mathrm{MD}} + 2\log\left(\frac{\hbar\omega}{k_B T}\right) + \log\frac{\exp\left(\frac{\hbar\omega}{k_B T}\right)}{\left[\exp\left(\frac{\hbar\omega}{k_B T}\right) - 1\right]^2}, \tag{S.3}$$

where $k_B$ is the Boltzmann constant, $\hbar$ is the Planck constant, $\omega$ is the frequency of molecular vibrations, and $T$ is the temperature. The bias is a monotonically decreasing function of frequency $\omega$, which is described as a black-box function of polymers with their molecular features. Hereafter, we consider the calibration of this systematic bias using the affine transfer model.

### B.3.1 Data

Experimental values of the specific heat capacity of the 70 polymers were collected from PoLyInfo [39]. The MD simulation was also applied to calculate their heat capacities. For models to predict the log-transformed heat capacity, a given polymer with its chemical structure was translated into the 190-dimensional force field descriptors, using RadonPy [38][2].

The force field descriptor represents the distribution of the ten different force field parameters ( $t \in \mathcal{T} = \{\mathrm{mass}, \sigma, \epsilon, \mathrm{charge}, r_0, K_{\mathrm{bond}}, \mathrm{polar}, \theta_0, K_{\mathrm{angle}}, K_{\mathrm{dih}}\}$ that make up the empirical potential (i.e., the General AMBER force field [37] version 2 (GAFF2)) of the classical MD simulation. The

---

[2]https://github.com/RadonPy/RadonPy

**Algorithm S.1** Block relaxation algorithm for the model in Eq. (S.4).

---
**Initialize:** $\alpha_0 \leftarrow \hat{\alpha}_{0,\text{olr}}, \alpha_1 \leftarrow \hat{\alpha}_{1,\text{olr}}, \beta \leftarrow 0, \boldsymbol{\gamma} \leftarrow \hat{\boldsymbol{\gamma}}_{\text{diff}}$
**repeat**
    $\boldsymbol{\alpha} \leftarrow \arg\min_{\boldsymbol{\alpha}} F_{\boldsymbol{\alpha},\beta,\boldsymbol{\gamma}}$
    $\beta \leftarrow \arg\min_{\beta} F_{\boldsymbol{\alpha},\beta,\boldsymbol{\gamma}}$
    $\boldsymbol{\gamma} \leftarrow \arg\min_{\boldsymbol{\gamma}} F_{\boldsymbol{\alpha},\beta,\boldsymbol{\gamma}}$
**until** convegence

---

detailed descriptions for each parameter are listed in Table S.1. For each $t$, pre-defined values are assigned to their constituent elements in a polymer, such as individual atoms (mass, charge, $\sigma$, and $\epsilon$), bonds ($r_0$, $K_{\text{bond}}$, and polar), angles ($\theta_0$ and $K_{\text{angle}}$), or dihedral angles ($K_{\text{dih}}$), respectively. The probability density function of the assigned values of $t$ is then estimated and discretized into 10 points corresponding to 10 different element species such as hydrogen and carbon for mass, and 20 equally spaced grid points for the other parameters. The details of the descriptor calculations are described in [40].

The source feature $f_s$ was given as the log-transformed value of $C_{\text{P}}^{\text{MD}}$. Therefore, $f_s$ is no longer a function of $\boldsymbol{x}$; this modeling was intended for calibrating the MD-calculated properties.

We randomly sampled 60 training polymers and tested the prediction performance of a trained model on the remaining 10 polymers 20 times. The PoLyInfo sample identifiers for the selected polymers are listed in the code.

### B.3.2 Model Definition and Hyperparameter Search

As described above, the 190-dimensional force field descriptor consists of ten blocks corresponding to different types of features. The $J_t$ features that make up block $t$ represent discretized values of the density function of the force field parameters assigned to the atoms, bonds, or dihedral angles that constitute the given polymer. Therefore, the regression coefficients of the features within a block should be estimated smoothly. To this end, we imposed fused regularization on the parameters as

$$\lambda_1 \|\boldsymbol{\gamma}\|_2^2 + \lambda_2 \sum_{t \in \mathcal{T}} \sum_{j=2}^{J_t} \left(\gamma_{t,j} - \gamma_{t,j-1}\right)^2,$$

where $\mathcal{T} = \{\text{mass}, \text{charge}, \epsilon, \sigma, K_{\text{bond}}, r_0, K_{\text{angle}}, \theta, K_{\text{dih}}\}$, and $J_t = 10$ for $t = \text{mass}$ and $J_t = 20$ otherwise. The regression coefficient $\gamma_{t,j}$ corresponds to the $j$-th feature of block $t$.

**Ordinary Linear Regression**    The experimental heat capacity $y = \log C_{\text{P}}^{\text{exp}}$ was regressed on the MD-calculated property, without regularization, as $\hat{y} = \alpha_0 + \alpha_1 f_s$ where $\hat{y}$ denotes the conditional expectation and $f_s = \log C_{\text{P}}^{\text{MD}}$.

**Learning the Log-Difference**    We calculated the log-difference $\log C_{\text{P}}^{\text{exp}} - \log C_{\text{P}}^{\text{MD}}$ and trained the linear model with the ridge penalty. The hyperparameters $\lambda_1$ and $\lambda_2$ for the scale- and smoothness-regularizers were determined based on 5-fold cross validation across 25 equally space grids in the interval $[10^{-2}, 10^2]$ for $\lambda_1$ and across the set $\{50, 100, 150\}$ for $\lambda_2$.

**Affine Transfer**    The log-transformed value of $C_{\text{p}}^{\text{exp}}$ is modeled as

$$y := \log C_{\text{P}}^{\text{exp}} = \underbrace{\alpha_0 + \alpha_1 f_s}_{g_1} - \underbrace{(\beta f_s + 1)}_{g_2} \cdot \underbrace{\boldsymbol{x}^{\top}\boldsymbol{\gamma}}_{g_3} + \epsilon_{\sigma}, \tag{S.4}$$

where $\epsilon_{\sigma}$ represents observation noise, and $\alpha_0, \alpha_1, \beta$ and $\boldsymbol{\gamma}$ are unknown parameters to be estimated. When $\alpha_1 = 1$ and $\beta = 0$, Eq. (S.4) is consistent with the theoretical equation in Eq. (S.3) in which the quantum effect is linearly modeled as $\alpha_0 + \boldsymbol{x}^{\top}\boldsymbol{\gamma}$.

Table S.2: Mean and standard deviation of RMSE of three prediction models.

| Model | RMSE (log J/kg·K) |
|---|---|
| $y = \alpha_0 + \alpha_1 f_s + \epsilon_\sigma$ | $0.1403 \pm 0.0461$ |
| $y = f_s + \boldsymbol{x}^\top \boldsymbol{\gamma} + \epsilon_\sigma$ | $0.1368 \pm 0.04265$ |
| $y = \alpha_0 + \alpha_1 f_s - (\beta f_s + 1)\boldsymbol{x}^\top \boldsymbol{\gamma} + \epsilon_\sigma$ | $\mathbf{0.1357 \pm 0.04173}$ |

In the model training, the objective function was given as follows:

$$F_{\boldsymbol{\alpha},\beta,\boldsymbol{\gamma}} = \frac{1}{n} \sum_{i=1}^n \left\{ y_i - (\alpha_0 + \alpha_1 f_{s,i} - (\beta f_{s,i} + 1)\boldsymbol{x}^\top \boldsymbol{\gamma}) \right\}^2$$

$$+ \lambda_\beta \beta^2 + \lambda_{\gamma,1} \|\boldsymbol{\gamma}\|_2^2 + \lambda_{\gamma,2} \sum_{t \in T} \sum_{j=2}^{J_t} \left( \gamma_{t,j} - \gamma_{t,j-1} \right)^2,$$

where $\boldsymbol{\alpha} = [\alpha_0 \ \alpha_1]^\top$. With a fixed $\lambda_\beta = 1$, the remaining hyperparameters $\lambda_{\gamma,1}$ and $\lambda_{\gamma,2}$ were optimized through 5-fold cross validation over 25 equally space grids in the interval $[10^{-2}, 10^2]$ for $\lambda_{\gamma,1}$ and across the set $\{50, 100, 150\}$ for $\lambda_{\gamma,2}$.

The algorithm to estimate the parameters $\boldsymbol{\alpha}$, $\beta$ and $\boldsymbol{\gamma}$ is described in Algorithm S.1, where $\alpha_{0,\text{olr}}$ and $\alpha_{1,\text{olr}}$ are the estimated parameters of the ordinary linear regression model, and $\hat{\gamma}_{\text{diff}}$ is the estimated parameter of the log-difference model. For each step, the full conditional minimization of $F_{\boldsymbol{\alpha},\beta,\boldsymbol{\gamma}}$ with respect to each parameter can be made analytically as

$$\arg \min_{\boldsymbol{\alpha}} F_{\boldsymbol{\alpha},\beta,\boldsymbol{\gamma}} = (F_s^\top F_s)^{-1} F_s^\top (\boldsymbol{y} + (\beta \boldsymbol{f}_{\boldsymbol{s},1:n} + 1) \circ (X\boldsymbol{\gamma})),$$

$$\arg \min_{\beta} F_{\boldsymbol{\alpha},\beta,\boldsymbol{\gamma}} = -(\boldsymbol{f}_{\boldsymbol{s},1:n}^\top \text{diag}(X\boldsymbol{\gamma})^2 \boldsymbol{f}_{\boldsymbol{s},1:n} + n\lambda_2)^{-1} \boldsymbol{f}_{\boldsymbol{s},1:n}^\top \text{diag}(X\boldsymbol{\gamma})(\boldsymbol{y} - F_s\boldsymbol{\alpha} + X\boldsymbol{\gamma}),$$

$$\arg \min_{\boldsymbol{\gamma}} F_{\boldsymbol{\alpha},\beta,\boldsymbol{\gamma}} = -(X^\top \text{diag}(\beta \boldsymbol{f}_{\boldsymbol{s},1:n} + 1)^2 X + \Lambda)^{-1} X^\top \text{diag}(\beta \boldsymbol{f}_{\boldsymbol{s},1:n} + 1)(\boldsymbol{y} - F_s\boldsymbol{\alpha}),$$

where $X$ denote the matrix in which the $i$-th row is $x_i$, $\boldsymbol{y} = [y_1 \cdots y_n]^\top$, $\boldsymbol{f}_{\boldsymbol{s},1:n} = [f_{s,1} \cdots f_{s,n}]^\top$, $F_s = [\boldsymbol{f}_{\boldsymbol{s},1:n} \ \mathbf{1}]$, and $d = 190$. $\Lambda$ is a matrix including the two regularization parameters $\lambda_{\gamma,1}$ and $\lambda_{\gamma,2}$ as

$$\Lambda = D^\top D, \text{ where } D = \begin{bmatrix} \lambda_{\gamma,1} I_d \\ \lambda_{\gamma,2} M \end{bmatrix}, M = \begin{bmatrix} -1 & 1 & 0 & \cdots & 0 & 0 \\ 0 & -1 & 1 & \cdots & 0 & 0 \\ \vdots & \vdots & \vdots & \ddots & \vdots & \vdots \\ 0 & 0 & 0 & \cdots & 0 & 0 \\ \vdots & \vdots & \vdots & \ddots & \vdots & \vdots \\ 0 & 0 & 0 & \cdots & -1 & 1 \end{bmatrix} \begin{matrix} \\ \\ \\ \leftarrow m\text{-th rows} \\ \\ \\ \end{matrix},$$

where $m \in \{10, 30, 50, 70, 90, 110, 130, 150, 170\}$. Note that the matrix $M$ is the same as the matrix $[\mathbf{0} \ I_{189}] - [I_{189} \ \mathbf{0}]$ except that the $m$-th row is all zeros. Note also that $M \in \mathbb{R}^{189 \times 190}$, and therefore $D \in \mathbb{R}^{279 \times 190}$ and $\Lambda \in \mathbb{R}^{190 \times 190}$.

The stopping criterion of the algorithm was set as

$$\max_{\theta \in \{\boldsymbol{\alpha},\beta,\boldsymbol{\gamma}\}} \frac{\max_i \left| \theta_i^{(\text{new})} - \theta_i^{(\text{old})} \right|}{\max_i \left| \theta_i^{(\text{old})} \right|} < 10^{-4}, \tag{S.5}$$

where $\theta_i$ denotes the $i$-th element of the parameter $\theta$. This convergence criterion is employed in several existing machine learning libraries, e.g., scikit-learn [3].

### B.3.3 Results

Table S.2 summarizes the prediction performance (RMSE) of the three models. The ordinary linear model $y = \alpha_0 + \alpha_1 f_s + \epsilon_\sigma$, which ignores the force field descriptors, exhibited the lowest prediction

---

[3] `https://scikit-learn.org/stable/modules/generated/sklearn.linear_model.Lasso.html`

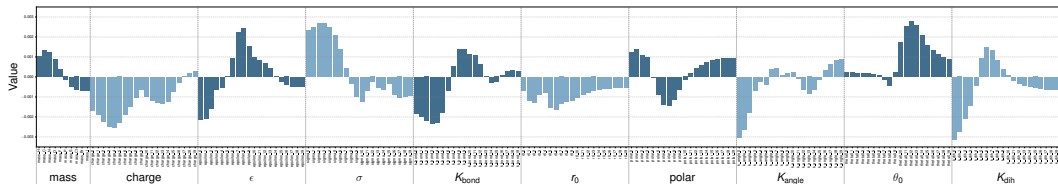

Figure S.5: Bar plot of regression coefficients $\gamma$ of linear calibrator filling the discrepancy between experimental and MD-calculated specific heat capacity of amorphous polymers.

performance. The other two calibration models $y = f_s + \boldsymbol{x}^\top \boldsymbol{\gamma} + \epsilon_\sigma$ and the full model in Eq. (S.4) reached almost the same accuracy, but the latter had achieved slightly better prediction accuracy. The estimated parameters of the full model were $\alpha_1 \approx 0.889$ and $\beta \approx -0.004$. The model form is highly consistent with the theoretical equation in Eq. (S.3) as well as the restricted model ($\alpha_1 = 1, \beta = 0$). This supports the validity of the theoretical model in [38].

It is expected that physicochemical insights can be obtained by examining the estimated coefficient $\boldsymbol{\gamma}$, which would capture the contribution of the force field parameters to the quantum effects. The magnitude of the quantum effect is a monotonically increasing function of the frequency $\omega$, and is known to be highly related to the descriptors $\epsilon$, $K_{\mathrm{dih}}$, $K_{\mathrm{bond}}$, $K_{\mathrm{angle}}$ and mass. According to physicochemical intuition, it is considered that as $\epsilon$, $K_{\mathrm{bond}}$, $K_{\mathrm{angle}}$, and $K_{\mathrm{dih}}$ decrease, their potential energy surface becomes shallow, which leads to the decrease of $\omega$, and in turn the decrease of quantum effects. Furthermore, because the molecular vibration of light-weight atoms is faster than that of heavy atoms, $\omega$ and quantum effects should theoretically increase with decreasing mass.

Figure S.5 shows the mean values of the estimated parameter $\boldsymbol{\gamma}$ for the full calibration model. The physical relationships described above can be captured consistently with the estimated coefficients. The coefficients in lower regions of $\epsilon$, $K_{\mathrm{bond}}$, $K_{\mathrm{angle}}$ and $K_{\mathrm{dih}}$ showed large negative values, indicating that polymers containing more atoms, bonds, angles, and dihedral angles with lower values will have smaller quantum effects. Conversely, the coefficients in lower regions of mass showed positive large values, meaning that polymers containing more atoms with smaller masses will have larger quantum effects. As illustrated in this example, separate inclusion of the domain-common and domain-specific factors in the affine transfer model enables us to infer the features relevant to the cross-domain differences.

## C  Experimental Details

Instructions for obtaining the datasets used in the experiments are described in the code.

### C.1  Kinematics of the Robot Arm

#### C.1.1  Data

We used the SARCOS dataset in [23]. The task is to predict the feed-forward torque required to follow the desired trajectory in the seven joints of the SARCOS anthropomorphic robot arm. The twenty one features representing the joints' position, velocity, and acceleration were used as $\boldsymbol{x}$. The observed values of six torques other than the torque at the joint in the target domain were given to the source features $\boldsymbol{f_s}$. The dataset includes 44,484 training samples and 4,449 test samples. We selected $\{5, 10, 15, 20, 30, 40, 50\}$ samples randomly from the training set. The prediction performances of the trained models were evaluated using the 4,449 test samples. Repeated experiments were conducted 20 times with different independently sampled datasets.

#### C.1.2  Model Definition and Hyperparameter Search

**Source model**  For each target task, a multi-task neural network was trained to predict the torque values of the remaining six source tasks. The source model shares four layers (256-128-64-32) up to the final layer, and only the output layer is task-specific. We used all training data and Adagrad [41] with learning rate of 0.01.

**Algorithm S.2** Block relaxation algorithm for **AffineTL-full**.

---

**Initialize:** $\boldsymbol{a}_0 \leftarrow (K_1 + \lambda_1 I_n)^{-1}\boldsymbol{y}$, $\boldsymbol{b}_0 \sim \mathcal{N}(\boldsymbol{0}, I_n)$, $\boldsymbol{c}_0 \sim \mathcal{N}(\boldsymbol{0}, I_n)$, $d_0 \leftarrow 0.5$
**repeat**
$\quad \boldsymbol{a} \leftarrow (K_1 + \lambda_1 I_n)^{-1}(\boldsymbol{y} - (K_2\boldsymbol{b} + \boldsymbol{1}) \circ (K_3\boldsymbol{c}) - d\boldsymbol{1})$
$\quad \boldsymbol{b} \leftarrow (\mathrm{Diag}(K_3\boldsymbol{c})^2 K_2 + \lambda_2 I_n)^{-1}((K_3\boldsymbol{c}) \circ (\boldsymbol{y} - K_1\boldsymbol{a} - K_3\boldsymbol{c} - d\boldsymbol{1}))$
$\quad \boldsymbol{c} \leftarrow (\mathrm{Diag}(K_2\boldsymbol{b} + \boldsymbol{1})^2 K_3 + \lambda_3 I_n)^{-1}((K_2\boldsymbol{b} + \boldsymbol{1}) \circ (\boldsymbol{y} - K_1\boldsymbol{a} - d\boldsymbol{1}))$
$\quad d \leftarrow \langle \boldsymbol{y} - K_1\boldsymbol{a} - (K_2\boldsymbol{b} + \boldsymbol{1}) \circ (K_3\boldsymbol{c}), \boldsymbol{1} \rangle / n$
**until** convergence

---

**Direct, Only source, Augmented, HTL-offset, HTL-scale** For each procedure, we used kernel ridge regression with the RBF kernel $k(\boldsymbol{x}, \boldsymbol{x}') = \exp(-\frac{1}{2\ell^2}\|\boldsymbol{x} - \boldsymbol{x}'\|_2^2)$. The scale parameter $\ell$ was set to the square root of the input dimension as $\ell = \sqrt{21}$ for **Direct**, **HTL-offset** and **HTL-scale**, $\ell = \sqrt{6}$ for **Only source** and $\ell = \sqrt{27}$ for **Augmented**. The regularization parameter $\lambda$ was selected in 5-fold cross-validation in which the grid search was performed over 50 grid points in the interval $[10^{-4}, 10^2]$.

**AffineTL-full, AffineTL-const** We considered the following kernels:

$$k_1(\boldsymbol{f_s}(\boldsymbol{x}), \boldsymbol{f_s}(\boldsymbol{x}')) = \exp\left(-\frac{1}{2\ell^2}\|\boldsymbol{f_s}(\boldsymbol{x}) - \boldsymbol{f_s}(\boldsymbol{x}')\|_2^2\right) \ (\ell = \sqrt{6}),$$

$$k_2(\boldsymbol{f_s}(\boldsymbol{x}), \boldsymbol{f_s}(\boldsymbol{x}')) = \exp\left(-\frac{1}{2\ell^2}\|\boldsymbol{f_s}(\boldsymbol{x}) - \boldsymbol{f_s}(\boldsymbol{x}')\|_2^2\right) \ (\ell = \sqrt{6}),$$

$$k_3(\boldsymbol{x}, \boldsymbol{x}') = \exp\left(-\frac{1}{2\ell^2}\|\boldsymbol{x} - \boldsymbol{x}'\|_2^2\right) \ (\ell = \sqrt{27}),$$

for $g_1, g_2$ and $g_3$ in the affine transfer model, respectively.

Hyperparameters to be optimized are the three regularization parameters $\lambda_1, \lambda_2$ and $\lambda_3$. We performed 5-fold cross-validation to identify the best hyperparameter set from the candidate points; $\{10^{-3}, 10^{-2}, 10^{-1}, 1\}$ for $\lambda_1$ and $\{10^{-2}, 10^{-1}, 1, 10\}$ for each of $\lambda_2$ and $\lambda_3$.

To learn the **AffineTL-full** and **AffineTL-const**, we used the following objective functions:

**AffineTL-full** $\|\boldsymbol{y} - (K_1\boldsymbol{a} + (K_2\boldsymbol{b} + \boldsymbol{1}) \circ (K_3\boldsymbol{c}) + d\boldsymbol{1})\|_2^2 + \lambda_1 \boldsymbol{a}^\top K_1 \boldsymbol{a} + \lambda_2 \boldsymbol{b}^\top K_2 \boldsymbol{b} + \lambda_3 \boldsymbol{c}^\top K_3 \boldsymbol{c},$

**AffineTL-const** $\dfrac{1}{n}\|\boldsymbol{y} - (K_1\boldsymbol{a} + K_3\boldsymbol{c} + d\boldsymbol{1})\|_2^2 + \lambda_1 \boldsymbol{a}^\top K_1 \boldsymbol{a} + \lambda_3 \boldsymbol{c}^\top K_3 \boldsymbol{c}.$

Algorithm S.2 summarizes the block relaxation algorithm for **AffineTL-full**. For **AffineTL-const**, we found the optimal parameters as follows:

$$\begin{bmatrix} \hat{\boldsymbol{a}} \\ \hat{\boldsymbol{c}} \\ \hat{d} \end{bmatrix} = \left(\begin{bmatrix} K_1 \\ K_3 \\ \boldsymbol{1}^\top \end{bmatrix}\begin{bmatrix} K_1 & K_3 & \boldsymbol{1} \end{bmatrix} + \begin{bmatrix} \lambda_1 K_1 & & \\ & \lambda_3 K_3 & \\ & & 0 \end{bmatrix}\right)^{-1}\begin{bmatrix} K_1 \\ K_3 \\ \boldsymbol{1}^\top \end{bmatrix}\boldsymbol{y}$$

The stopping criterion for the algorithm was the same as Eq. (S.5).

**Fine-tuning** The target network was constructed by adding a one-dimensional output layer to the shared layers of the source network. As initial values for the training, we used the weights of the source neural network for the shared layer and the average of the multidimensional output layer of the source network for the output layer. Adagrad [41] was used for the optimization. The learning rate was fixed at 0.01 and the number of training epochs was selected from $\{1, 2, 3, 4, 5, 6, 7, 8, 9, 10, 20, 50, 100\}$ through 5-fold cross-validation.

**MAML** A fully connected neural network with 256-64-32-16-1 layer width was used, and the initial values were searched through MAML [25] using the six source tasks. The obtained base model was fine-tuned with the target samples. Adam [42] with a fixed learning rate of 0.01 was used for the optimization. The number of training epochs was selected from $\{1, 2, 3, 4, 5, 6, 7, 8, 9, 10, 20, 50, 100\}$ through 5-fold cross-validation.

$L^2$**-SP**  $L^2$-SP is a regularization method proposed by [26] in which the following regularization term is added so that the weights of the target network are estimated in the neighborhood of the weights of the source network:

$$\Omega(\boldsymbol{w}) = \frac{\lambda}{2}\|\boldsymbol{w} - \boldsymbol{w_s}\|_2^2, \tag{S.6}$$

where $\boldsymbol{w}$ and $\boldsymbol{w_s}$ are the weights of the target and source model, respectively, and $\lambda > 0$ is a hyperparameter. We used the weights of the source network as the initial point for the training of the target model, and added a regularization parameter as in Eq. (S.6). Adagrad [41] were used for the optimizer, and the regularization parameters and learning rate were fixed at $0.01$ and $0.001$, respectively. The number of training epochs was selected from $\{1, 2, 3, 4, 5, 6, 7, 8, 9, 10, 20, 50, 100\}$ through 5-fold cross-validation.

**PAC-Net**  PAC-Net, proposed in [27], is a TL method that leverages pruning of the weights of the source network. Its training strategy consists of three steps: identifying the important weights in the source model, fine-tuning them using the source samples, and updating the remaining weights using the target samples.

Firstly, we pruned the bottom $10\%$ of weights, based on absolute value, from the pre-trained source network. Following this, the remaining weights were retrained using the stochastic gradient descent (SGD). Finally, the pruned weights were retrained using target samples. For the final training phase, SGD with learning rate $0.01$, was employed, and the number of training epochs was selected from $\{1, 2, 3, 4, 5, 6, 7, 8, 9, 10, 20, 50, 100\}$ through 5-fold cross-validation.

Table S.3: Performance on predicting the torque values at the first and seventh joints of the SARCOS robot arm. Definitions of an asterisk and bold type are the same as in Table 1.

| Target | Model | Number of training samples | | | | | | |
| | | n < d | | | n ≈ d | n > d | | |
| | | 5 | 10 | 15 | 20 | 30 | 40 | 50 |
|---|---|---|---|---|---|---|---|---|
| Torque 1 | Direct | 21.3 ± 2.04 | 18.9 ± 2.11 | **17.4 ± 1.79** | 15.8 ± 1.70 | 13.7 ± 1.26 | 12.2 ± 1.61 | 10.8 ± 1.23 |
| | Only source | 24.0 ± 6.37 | 22.3 ± 3.10 | 21.0 ± 2.49 | 19.7 ± 1.34 | 18.5 ± 1.92 | 17.6 ± 1.59 | 17.3 ± 1.31 |
| | Augmented | 21.8 ± 2.88 | 19.2 ± 1.37 | 17.8 ± 2.30 | **15.7 ± 1.53** | **13.3 ± 1.19** | **11.9 ± 1.37** | **10.7 ± 0.954** |
| | HTL-offset | 23.7 ± 6.50 | 21.2 ± 3.85 | 19.8 ± 3.23 | 17.8 ± 2.35 | 16.2 ± 3.31 | 15.0 ± 3.16 | 15.1 ± 2.76 |
| | HTL-scale | 23.3 ± 4.47 | 22.1 ± 5.31 | 20.4 ± 3.84 | 18.5 ± 2.72 | 17.6 ± 2.41 | 16.9 ± 2.10 | 16.7 ± 1.74 |
| | AffineTL-full | **21.2 ± 2.23** | **18.8 ± 1.31** | 18.6 ± 2.83 | 15.9 ± 1.65 | 13.7 ± 1.53 | 12.3 ± 1.45 | 11.1 ± 1.12 |
| | AffineTL-const | 21.2 ± 2.21 | 18.8 ± 1.44 | 17.7 ± 2.44 | 15.9 ± 1.58 | 13.4 ± 1.15 | 12.2 ± 1.54 | 10.9 ± 1.02 |
| | Fine-tune | 25.0 ± 7.11 | 20.5 ± 3.33 | 18.6 ± 2.10 | 17.6 ± 2.55 | 14.1 ± 1.39 | 12.6 ± 1.13 | 11.1 ± 1.03 |
| | MAML | 29.8 ± 12.3 | 22.5 ± 3.21 | 20.8 ± 2.12 | 20.3 ± 3.14 | 16.7 ± 3.00 | 14.4 ± 1.85 | 13.4 ± 1.19 |
| | $L^2$-SP | 24.9 ± 7.09 | 20.5 ± 3.30 | 18.8 ± 2.04 | 18.0 ± 2.45 | 14.5 ± 1.36 | 13.0 ± 1.13 | 11.6 ± 0.983 |
| | PAC-Net | 25.2 ± 8.68 | 22.7 ± 5.60 | 20.7 ± 2.65 | 20.1 ± 2.16 | 18.5 ± 2.77 | 17.6 ± 1.85 | 17.1 ± 1.38 |
| Torque 2 | Direct | 15.8 ± 2.37 | 13.0 ± 1.41 | 11.5 ± 0.985 | 10.4 ± 0.845 | 9.20 ± 0.827 | 8.35 ± 0.802 | 7.78 ± 0.780 |
| | Only source | 14.9 ± 1.77 | 13.6 ± 2.51 | 12.3 ± 1.77 | 11.2 ± 1.16 | 10.6 ± 1.22 | 9.74 ± 0.920 | 9.06 ± 0.785 |
| | Augmented | 15.2 ± 1.95 | **12.3 ± 0.923** | **11.4 ± 1.48** | **10.2 ± 0.813** | **9.07 ± 0.983** | **8.06 ± 0.862** | **7.23 ± 0.629** |
| | HTL-offset | 14.8 ± 1.71 | 13.4 ± 2.41 | 12.2 ± 1.81 | 10.9 ± 1.29 | 10.4 ± 1.37 | 9.32 ± 1.11 | 8.78 ± 0.829 |
| | HTL-scale | 14.8 ± 1.71 | 13.4 ± 2.47 | 12.2 ± 1.82 | 11.0 ± 1.32 | 10.5 ± 1.28 | 9.39 ± 1.01 | 8.91 ± 0.946 |
| | AffineTL-full | 14.7 ± 1.83 | 13.0 ± 1.34 | 11.9 ± 1.22 | 11.3 ± 1.39 | 9.38 ± 0.842 | 8.25 ± 0.932 | 7.34 ± 0.605 |
| | AffineTL-const | **14.6 ± 1.47** | 12.6 ± 1.09 | 11.5 ± 0.807 | 10.5 ± 1.19 | 9.28 ± 0.828 | 8.35 ± 1.06 | 7.33 ± 0.57 |
| | Fine-tune | 24.4 ± 5.87 | 15.0 ± 2.01 | 13.6 ± 2.31 | 11.9 ± 1.21 | 10.7 ± 0.897 | 9.52 ± 0.774 | 8.43 ± 0.907 |
| | MAML | 21.8 ± 7.33 | 14.8 ± 4.51 | 13.1 ± 2.69 | 11.5 ± 2.24 | 9.77 ± 1.24 | 8.90 ± 1.10 | 7.89 ± 0.713 |
| | $L^2$-SP | 24.4 ± 5.87 | 15.1 ± 2.02 | 13.6 ± 2.29 | 12.0 ± 1.22 | 10.8 ± 0.886 | 9.70 ± 0.78 | 8.68 ± 0.868 |
| | PAC-Net | 24.0 ± 6.94 | 16.7 ± 4.14 | 13.7 ± 2.36 | 13.2 ± 2.49 | 12.4 ± 2.05 | 11.6 ± 0.844 | 11.2 ± 0.706 |
| Torque 3 | Direct | 9.91 ± 1.65 | 8.15 ± 1.01 | 7.39 ± 1.21 | 6.84 ± 0.878 | 5.90 ± 0.850 | 5.26 ± 0.774 | 4.66 ± 0.523 |
| | Only source | 9.00 ± 1.44 | 7.51 ± 1.05 | 6.90 ± 1.15 | 6.51 ± 0.930 | 5.67 ± 0.890 | 5.29 ± 0.840 | 4.89 ± 0.604 |
| | Augmented | 9.47 ± 1.35 | 7.72 ± 1.05 | 6.99 ± 1.25 | 6.29 ± 0.967 | 5.42 ± 0.938 | **4.76 ± 0.826** | **4.32 ± 0.592** |
| | HTL-offset | **8.96 ± 1.42** | 7.47 ± 1.06 | 6.88 ± 1.15 | 6.39 ± 0.952 | 5.58 ± 0.856 | 5.18 ± 0.821 | 4.83 ± 0.603 |
| | HTL-scale | 9.05 ± 1.40 | 7.49 ± 1.08 | 6.89 ± 1.18 | 6.63 ± 1.03 | 5.60 ± 0.955 | 5.21 ± 0.836 | 4.86 ± 0.503 |
| | AffineTL-full | 9.24 ± 1.46 | **7.45 ± 1.25** | 6.85 ± 1.23 | 6.28 ± 0.930 | 5.54 ± 1.15 | 4.89 ± 0.907 | 4.46 ± 0.733 |
| | AffineTL-const | 9.08 ± 1.21 | 7.55 ± 0.974 | **6.67 ± 1.00** | **6.17 ± 0.916** | **5.42 ± 0.971** | 4.85 ± 0.752 | 4.42 ± 0.614 |
| | Fine-tune | 9.00 ± 2.14 | 7.38 ± 1.09 | 6.72 ± 1.01 | *5.91 ± 0.734 | *5.26 ± 0.541 | 4.86 ± 0.488 | 4.41 ± 0.325 |
| | MAML | 9.50 ± 4.94 | *7.11 ± 0.966 | *6.44 ± 1.01 | *5.92 ± 0.793 | *5.22 ± 0.626 | 4.87 ± 0.539 | 4.79 ± 0.525 |
| | $L^2$-SP | 9.00 ± 2.14 | 7.39 ± 1.08 | 6.73 ± 1.02 | *5.91 ± 0.73 | 5.39 ± 0.633 | 4.89 ± 0.493 | 4.46 ± 0.319 |
| | PAC-Net | 9.14 ± 2.11 | *7.31 ± 1.03 | *6.33 ± 0.841 | *5.96 ± 0.926 | 5.34 ± 0.633 | 5.17 ± 0.474 | 5.05 ± 0.371 |
| Torque 4 | Direct | 14.2 ± 2.30 | 11.1 ± 2.28 | 9.49 ± 2.19 | 7.78 ± 1.02 | 6.86 ± 0.768 | 6.13 ± 0.714 | 5.48 ± 0.592 |
| | Only source | 13.1 ± 3.36 | 9.62 ± 2.05 | 8.38 ± 2.06 | 7.06 ± 1.32 | 6.36 ± 1.24 | 5.79 ± 0.768 | 5.37 ± 0.897 |
| | Augmented | 13.5 ± 2.83 | 9.69 ± 1.89 | 8.51 ± 1.84 | *6.96 ± 1.03 | *6.09 ± 0.931 | *5.39 ± 0.685 | *4.87 ± 0.618 |
| | HTL-offset | 13 ± 3.38 | 9.62 ± 2.05 | 8.34 ± 2.00 | 7.02 ± 1.24 | 6.26 ± 1.17 | 5.76 ± 0.764 | 5.36 ± 0.897 |
| | HTL-scale | 13.0 ± 3.35 | 9.63 ± 2.07 | **8.30 ± 1.95** | 7.01 ± 1.16 | 6.30 ± 1.17 | 5.77 ± 0.758 | 5.37 ± 0.902 |
| | AffineTL-full | 13.0 ± 2.69 | 9.48 ± 2.10 | 8.38 ± 1.85 | 7.14 ± 1.62 | *5.91 ± 0.838 | *5.45 ± 0.777 | *4.94 ± 0.603 |
| | AffineTL-const | 13.2 ± 3.16 | **9.32 ± 1.99** | 8.39 ± 1.84 | *6.88 ± 1.00 | *5.85 ± 0.710 | *5.55 ± 0.679 | *4.94 ± 0.581 |
| | Fine-tune | *11.7 ± 2.70 | *8.24 ± 1.31 | *6.71 ± 1.02 | *5.90 ± 0.971 | *5.17 ± 0.785 | *4.59 ± 0.442 | *4.21 ± 0.376 |
| | MAML | 14.3 ± 7.75 | 10.9 ± 3.44 | 9.55 ± 1.99 | 9.41 ± 2.33 | 7.98 ± 2.36 | 6.70 ± 1.25 | 6.18 ± 1.35 |
| | $L^2$-SP | *11.7 ± 2.70 | *8.24 ± 1.31 | *6.73 ± 1.01 | *5.92 ± 0.959 | *5.22 ± 0.765 | *4.67 ± 0.45 | *4.28 ± 0.363 |
| | PAC-Net | 11.2 ± 5.24 | *8.84 ± 2.75 | *7.64 ± 1.17 | 7.34 ± 1.56 | 6.77 ± 0.966 | 6.29 ± 0.536 | 6.02 ± 0.446 |
| Torque 5 | Direct | 1.07 ± 0.157 | 0.993 ± 0.0903 | 0.910 ± 0.119 | 0.847 ± 0.129 | 0.744 ± 0.113 | 0.686 ± 0.0996 | 0.623 ± 0.0944 |
| | Only source | 1.15 ± 0.214 | 1.04 ± 0.0775 | 0.998 ± 0.145 | 0.975 ± 0.133 | 0.863 ± 0.111 | 0.826 ± 0.155 | 0.775 ± 0.106 |
| | Augmented | **1.04 ± 0.113** | 0.987 ± 0.109 | 0.907 ± 0.120 | 0.874 ± 0.136 | 0.755 ± 0.130 | 0.710 ± 0.110 | 0.637 ± 0.0893 |
| | HTL-offset | 1.14 ± 0.221 | 1.02 ± 0.0864 | 0.965 ± 0.157 | 0.925 ± 0.141 | 0.837 ± 0.104 | 0.800 ± 0.156 | 0.738 ± 0.101 |
| | HTL-scale | 1.13 ± 0.194 | 1.01 ± 0.0786 | 0.980 ± 0.177 | 0.914 ± 0.132 | 0.830 ± 0.114 | 0.844 ± 0.171 | 0.785 ± 0.123 |
| | AffineTL-full | 1.04 ± 0.121 | 0.989 ± 0.175 | 0.907 ± 0.162 | 0.860 ± 0.170 | 0.747 ± 0.117 | 0.691 ± 0.0924 | 0.654 ± 0.0716 |
| | AffineTL-const | 1.05 ± 0.106 | **0.974 ± 0.102** | **0.899 ± 0.123** | 0.854 ± 0.121 | 0.756 ± 0.106 | 0.700 ± 0.0869 | 0.638 ± 0.0796 |
| | Fine-tune | 1.22 ± 0.356 | 1.04 ± 0.105 | 0.976 ± 0.0878 | 0.913 ± 0.137 | 0.749 ± 0.111 | 0.688 ± 0.103 | 0.598 ± 0.0697 |
| | MAML | 1.45 ± 0.479 | 1.18 ± 0.183 | 1.07 ± 0.208 | 0.999 ± 0.193 | 0.816 ± 0.211 | 0.703 ± 0.124 | 0.613 ± 0.0634 |
| | $L^2$-SP | 1.22 ± 0.355 | 1.04 ± 0.105 | 0.973 ± 0.0873 | 0.917 ± 0.133 | 0.756 ± 0.109 | 0.699 ± 0.113 | 0.606 ± 0.0644 |
| | PAC-Net | 1.27 ± 0.319 | 1.10 ± 0.115 | 1.03 ± 0.108 | 0.985 ± 0.145 | 0.881 ± 0.151 | 0.806 ± 0.118 | 0.781 ± 0.124 |
| Torque 6 | Direct | 1.86 ± 0.248 | 1.67 ± 0.192 | **1.50 ± 0.162** | **1.36 ± 0.159** | **1.22 ± 0.163** | 1.12 ± 0.102 | 1.05 ± 0.0916 |
| | Only source | 1.91 ± 0.230 | 1.87 ± 0.357 | 1.76 ± 0.179 | 1.64 ± 0.190 | 1.53 ± 0.255 | 1.36 ± 0.153 | 1.26 ± 0.0883 |
| | Augmented | 1.86 ± 0.180 | **1.66 ± 0.17** | 1.55 ± 0.219 | 1.45 ± 0.231 | 1.27 ± 0.265 | **1.11 ± 0.130** | **1.01 ± 0.0944** |
| | HTL-offset | 1.88 ± 0.214 | 1.81 ± 0.369 | 1.67 ± 0.246 | 1.54 ± 0.239 | 1.46 ± 0.267 | 1.33 ± 0.142 | 1.21 ± 0.133 |
| | HTL-scale | 2.05 ± 0.649 | 1.88 ± 0.389 | 1.71 ± 0.241 | 1.60 ± 0.308 | 1.62 ± 0.474 | 1.37 ± 0.158 | 1.25 ± 0.0922 |
| | AffineTL-full | **1.82 ± 0.232** | 1.74 ± 0.209 | 1.55 ± 0.242 | 1.43 ± 0.231 | 1.27 ± 0.230 | 1.13 ± 0.122 | 1.06 ± 0.191 |
| | AffineTL-const | 1.83 ± 0.173 | 1.68 ± 0.207 | 1.55 ± 0.236 | 1.40 ± 0.227 | 1.23 ± 0.198 | 1.12 ± 0.113 | 1.02 ± 0.0953 |
| | Fine-tune | 2.41 ± 0.375 | 2.03 ± 0.387 | 1.71 ± 0.463 | 1.49 ± 0.297 | 1.27 ± 0.257 | 1.15 ± 0.110 | 1.07 ± 0.0817 |
| | MAML | 2.69 ± 0.676 | 2.17 ± 0.511 | 1.96 ± 0.526 | 1.68 ± 0.373 | 1.42 ± 0.365 | 1.23 ± 0.111 | 1.16 ± 0.0829 |
| | $L^2$-SP | 2.41 ± 0.375 | 2.03 ± 0.371 | 1.72 ± 0.455 | 1.49 ± 0.299 | 1.28 ± 0.254 | 1.16 ± 0.108 | 1.09 ± 0.0777 |
| | PAC-Net | 2.47 ± 0.385 | 2.22 ± 0.550 | 2.22 ± 0.599 | 1.99 ± 0.372 | 1.91 ± 0.355 | 1.74 ± 0.144 | 1.69 ± 0.0595 |
| Torque 7 | Direct | 2.66 ± 0.307 | 2.13 ± 0.420 | 1.85 ± 0.418 | 1.54 ± 0.353 | 1.32 ± 0.200 | 1.18 ± 0.138 | 1.05 ± 0.111 |
| | Only source | 2.31 ± 0.618 | *1.73 ± 0.560 | *1.49 ± 0.513 | *1.22 ± 0.269 | *1.09 ± 0.232 | *0.969 ± 0.144 | *0.927 ± 0.170 |
| | Augmented | 2.47 ± 0.406 | 1.90 ± 0.515 | 1.67 ± 0.552 | *1.31 ± 0.214 | 1.16 ± 0.225 | *0.984 ± 0.149 | *0.897 ± 0.138 |
| | HTL-offset | 2.29 ± 0.621 | **1.69 ± 0.507** | 1.49 ± 0.513 | *1.22 ± 0.269 | *1.09 ± 0.233 | *0.969 ± 0.144 | *0.925 ± 0.171 |
| | HTL-scale | 2.32 ± 0.599 | *1.71 ± 0.516 | 1.51 ± 0.513 | *1.24 ± 0.271 | *1.12 ± 0.234 | *0.999 ± 0.175 | 0.948 ± 0.172 |
| | AffineTL-full | *2.23 ± 0.554 | *1.71 ± 0.501 | **1.45 ± 0.458** | *1.21 ± 0.256 | *1.06 ± 0.219 | *0.974 ± 0.164 | *0.870 ± 0.121** |
| | AffineTL-const | *2.30 ± 0.565 | *1.73 ± 0.420 | *1.48 ± 0.527 | *1.20 ± 0.243 | *1.04 ± 0.217 | *0.963 ± 0.161 | *0.884 ± 0.136 |
| | Fine-tune | *2.33 ± 0.511 | *1.62 ± 0.347 | *1.35 ± 0.340 | *1.12 ± 0.165 | *0.959 ± 0.12 | *0.848 ± 0.0824 | *0.790 ± 0.0547 |
| | MAML | 2.54 ± 1.29 | 1.90 ± 0.507 | 1.67 ± 0.313 | 1.63 ± 0.282 | 1.28 ± 0.272 | 1.20 ± 0.199 | 1.06 ± 0.111 |
| | $L^2$-SP | *2.33 ± 0.509 | *1.65 ± 0.378 | *1.35 ± 0.340 | *1.12 ± 0.165 | *0.968 ± 0.114 | *0.858 ± 0.0818 | *0.802 ± 0.0535 |
| | PAC-Net | 2.24 ± 0.706 | *1.61 ± 0.394 | *1.43 ± 0.389 | *1.24 ± 0.177 | *1.18 ± 0.100 | 1.13 ± 0.0726 | 1.100 ± 0.0589 |

## C.2 Evaluation of Scientific Papers

### C.2.1 Data

We used SciRepEval benchmark dataset for scientific documents, proposed in [23]. This dataset comprises abstracts, review scores, and decision statuses of papers submitted to various machine learning conferences. We conducted two primary tasks: predicting the average review score (a regression task) and determining the acceptance or rejection status of each paper (a binary classification task). The original input $x$, was represented as a two-gram bag-of-words vector derived from the abstract. As for source features $f_s$, we employed text embeddings from the abstracts, which were generated by various pre-trained language models, including BERT [28], SciBERT [29], T5 [30], and GPT-3 [31]. When building the vocabulary for the bag-of-words, we ignore phrases with document frequencies strictly higher than 0.9 or strictly lower than 0.01. Additionally, we eliminated certain stop-words using the default settings in scikit-learn [43]. The sentences containing URLs were removed from the abstracts because accepted papers tend to include GitHub links in their abstracts after acceptance, which may cause leakage of information to the prediction. The models were trained on a dataset comprising 8,166 instances, and their performance were subsequently evaluated on a test dataset of 2,043 instances.

### C.2.2 Model Definition and Hyperparamter Search

For both the affine model transfer and feature extraction, we employed neural networks with ReLU activation and dropout layer with 0.1 dropout rate. The parameters were optimized using Adagrad [41] with 0.01 learning rate.

**Affine Model Transfer** For functions $g_1$ and $g_2$ in the affine model transfer, we used a neural network composed of layers with widths 128, 64, 32, and 16, wherein the source features $f_s$ were used as inputs. The number of layers of each width was determined based on Bayesian optimization. Sigmoid activation was employed to the output of $g_2$ in order to facilitate the interpretation of $g_3$. For $g_3$, we employed a linear model with the input $x$. To prevent overfitting and promote model simplicity, we applied $\ell_1$ regularization to the parameters of $g_3$ with a regularization parameter of 0.01. The final output of the model was computed as $g_1 + g_2 \cdot g_3$. In the case of the binary classification task, we applied sigmoid activation function to this final output.

**Feature Extraction** As in the affine model transfer, we used a neural network composed of layers with widths 128, 64, 32, and 16, wherein the source features $f_s$ were used as inputs. The number of layers of each width was determined based on Bayesian optimization. In the case of the binary classification task, we applied sigmoid activation function to the final output.

## D  Proofs

### D.1  Proof of Theorem 2.4

*Proof.* According to Assumption 2.3, it holds that for any $p_t(y|\boldsymbol{x})$,

$$\psi_{f_s}\left(\int \phi_{f_s}(y)p_t(y|\boldsymbol{x})\mathrm{d}y\right) = \int y p_t(y|\boldsymbol{x})\mathrm{d}y. \tag{S.7}$$

(i) Let $\delta_{y_0}$ be the Dirac delta function supported on $y_0$. Substituting $p_t(y|\boldsymbol{x}) = \delta_{y_0}$ into Eq. (S.7), we have

$$\psi_{f_s}(\phi_{f_s}(y_0)) = y_0 \ (\forall y_0 \in \mathcal{Y}).$$

Under Assumption 2.3, this implies the property (i).

(ii) For simplicity, we assume the inputs $\boldsymbol{x}$ are fixed and $p_t(y|\boldsymbol{x}) > 0$. Applying the property (i) to Eq. (S.7) yields

$$\int \phi_{f_s}(y)p_t(y|\boldsymbol{x})\mathrm{d}y = \phi_{f_s}\left(\int y p_t(y|\boldsymbol{x})\mathrm{d}y\right).$$

We consider a two-component mixture $p_t(y|\boldsymbol{x}) = (1 - \epsilon)q(y|\boldsymbol{x}) + \epsilon h(y|\boldsymbol{x})$ with a mixing rate $\epsilon \in [0, 1]$, where $q$ and $h$ denote arbitrary probability density functions. Then, we have

$$\int \phi_{f_s}(y)\{(1 - \epsilon)q(y|\boldsymbol{x}) + \epsilon h(y|\boldsymbol{x})\}\mathrm{d}y = \phi_{f_s}\left(\int y\{(1 - \epsilon)q(y|\boldsymbol{x}) + \epsilon h(y|\boldsymbol{x})\}\mathrm{d}y\right).$$

Taking the derivative at $\epsilon = 0$, we have

$$\int \phi_{f_s}(y)\{h(y|\boldsymbol{x}) - q(y|\boldsymbol{x})\}\mathrm{d}y = \phi'_{f_s}\left(\int yq(y|\boldsymbol{x})\mathrm{d}y\right)\left(\int y\{h(y|\boldsymbol{x}) - q(y|\boldsymbol{x})\}\mathrm{d}y\right),$$

which yields

$$\int \{h(y|\boldsymbol{x}) - q(y|\boldsymbol{x})\}\{\phi_{f_s}(y) - \phi'_{f_s}(\mathbb{E}_q[Y|\boldsymbol{X} = \boldsymbol{x}])y\}\mathrm{d}y = 0. \tag{S.8}$$

For Eq. (S.8) to hold for any $q$ and $h$, $\phi_{f_s}(y) - \phi'_{f_s}(\mathbb{E}_q[Y|\boldsymbol{X} = \boldsymbol{x}])y$ must be independent of $y$. Thus, the function $\phi_{f_s}$ and its inverse $\psi_{f_s} = \phi_{f_s}^{-1}$ are limited to affine transformations with respect to $y$. Since $\phi$ depends on $y$ and $\boldsymbol{f_s}(\boldsymbol{x})$, it takes the form $\phi(y, \boldsymbol{f_s}(\boldsymbol{x})) = g_1(\boldsymbol{f_s}(\boldsymbol{x})) + g_2(\boldsymbol{f_s}(\boldsymbol{x}))y$. □

### D.2 Proof of Theorem 4.1

To bound the generalization error, we use the empirical and population Rademacher complexity $\hat{\mathfrak{R}}_S(\mathcal{F})$ and $\mathfrak{R}(\mathcal{F})$ of hypothesis class $\mathcal{F}$, defined as:

$$\hat{\mathfrak{R}}_S(\mathcal{F}) = \mathbb{E}_\sigma \sup_{f \in \mathcal{F}} \frac{1}{n} \sum_{i=1}^n \sigma_i f(\boldsymbol{x}_i), \qquad \mathfrak{R}(\mathcal{F}) = \mathbb{E}_S \hat{\mathfrak{R}}_S(\mathcal{F}),$$

where $\{\sigma_i\}_{i=1}^n$ is a set of Rademacher variables that are independently distributed and each take one of the values in $\{-1, +1\}$ with equal probability, and $S$ denotes a set of samples. The following proof is based on the one of Theorem 11 shown in [11].

*Proof of Theorem 4.1.* For any hypothesis class $\mathcal{F}$ with feature map $\Phi$ where $\|\Phi\|^2 \leq 1$, the following inequality holds:

$$\mathbb{E}_\sigma \sup_{\|\boldsymbol{\theta}\|^2 \leq \Lambda} \frac{1}{n} \sum_{i=1}^n \sigma_i \langle \boldsymbol{\theta}, \Phi(\boldsymbol{x}_i) \rangle \leq \sqrt{\frac{\Lambda}{n}}.$$

The proof is given, for example, in Theorem 6.12 of [44]. Thus, the empirical Rademacher complexity of $\tilde{\mathcal{H}}$ is bounded as

$$\hat{\mathfrak{R}}_S(\tilde{\mathcal{H}}) = \mathbb{E}_\sigma \sup_{\substack{\|\boldsymbol{\alpha}\|_{\mathcal{H}_1}^2 \le \lambda_{\boldsymbol{\alpha}}^{-1}\hat{R}_s, \\ \|\boldsymbol{\beta}\|_{\mathcal{H}_2}^2 \le \lambda_{\boldsymbol{\beta}}^{-1}\hat{R}_s, \\ \|\boldsymbol{\gamma}\|_{\mathcal{H}_3}^2 \le \lambda_{\boldsymbol{\gamma}}^{-1}\hat{R}_s}} \frac{1}{n}\sum_{i=1}^n \sigma_i \left\{ \langle \boldsymbol{\alpha}, \Phi_1(\boldsymbol{f_s}(\boldsymbol{x}_i)) \rangle_{\mathcal{H}_1} + \langle \boldsymbol{\beta}, \Phi_2(\boldsymbol{f_s}(\boldsymbol{x}_i)) \rangle_{\mathcal{H}_2} \langle \boldsymbol{\gamma}, \Phi(\boldsymbol{x}_i) \rangle_{\mathcal{H}_3} \right\}$$

$$\le \mathbb{E}_\sigma \sup_{\|\boldsymbol{\alpha}\|_{\mathcal{H}_1}^2 \le \lambda_{\boldsymbol{\alpha}}^{-1}\hat{R}_s} \frac{1}{n}\sum_{i=1}^n \sigma_i \langle \boldsymbol{\alpha}, \Phi_1(\boldsymbol{f_s}(\boldsymbol{x}_i)) \rangle_{\mathcal{H}_1}$$

$$+ \sup_{\substack{\|\boldsymbol{\beta}\|_{\mathcal{H}_2}^2 \le \lambda_{\boldsymbol{\beta}}^{-1}\hat{R}_s, \\ \|\boldsymbol{\gamma}\|_{\mathcal{H}_3}^2 \le \lambda_{\boldsymbol{\gamma}}^{-1}\hat{R}_s}} \frac{1}{n}\sum_{i=1}^n \sigma_i \langle \boldsymbol{\beta} \otimes \boldsymbol{\gamma}, \Phi_2(\boldsymbol{f_s}(\boldsymbol{x}_i)) \otimes \Phi(\boldsymbol{x}_i) \rangle_{\mathcal{H}_2 \otimes \mathcal{H}_3}$$

$$\le \mathbb{E}_\sigma \sup_{\|\boldsymbol{\alpha}\|_{\mathcal{H}_1}^2 \le \lambda_{\boldsymbol{\alpha}}^{-1}\hat{R}_s} \frac{1}{n}\sum_{i=1}^n \sigma_i \langle \boldsymbol{\alpha}, \Phi_1(\boldsymbol{f_s}(\boldsymbol{x}_i)) \rangle_{\mathcal{H}_1}$$

$$+ \sup_{\|\boldsymbol{\beta} \otimes \boldsymbol{\gamma}\|_{\mathcal{H}_2 \otimes \mathcal{H}_3}^2 \le \lambda_{\boldsymbol{\beta}}^{-1}\lambda_{\boldsymbol{\gamma}}^{-1}\hat{R}_s^2} \frac{1}{n}\sum_{i=1}^n \sigma_i \langle \boldsymbol{\beta} \otimes \boldsymbol{\gamma}, \Phi_2(\boldsymbol{f_s}(\boldsymbol{x}_i)) \otimes \Phi(\boldsymbol{x}_i) \rangle_{\mathcal{H}_2 \otimes \mathcal{H}_3}$$

$$\le \sqrt{\frac{\hat{R}_s}{\lambda_{\boldsymbol{\alpha}} n}} + \sqrt{\frac{\hat{R}_s^2}{\lambda_{\boldsymbol{\beta}} \lambda_{\boldsymbol{\gamma}} n}} \tag{S.9}$$

$$\le \sqrt{\frac{\hat{R}_s}{n}} \left\{ \sqrt{\frac{1}{\lambda_{\boldsymbol{\alpha}}}} + \sqrt{\frac{L}{\lambda_{\boldsymbol{\beta}} \lambda_{\boldsymbol{\gamma}}}} \right\}.$$

The first inequality follows from the subadditivity of supremum. The last inequality follows from the fact that $\hat{R}_s \le P_n \ell(y, \langle 0, \Phi_1 \rangle) + \lambda_{\boldsymbol{\alpha}} \|0\|^2 \le L$.

Let $C = \sqrt{\frac{1}{\lambda_{\boldsymbol{\alpha}}}} + \sqrt{\frac{L}{\lambda_{\boldsymbol{\beta}} \lambda_{\boldsymbol{\gamma}}}}$, and applying Talagrand's lemma [44] and Jensen's inequality, we obtain

$$\mathfrak{R}(\mathcal{L}) = \mathbb{E}\hat{\mathfrak{R}}_S(\mathcal{L}) \le \mu_\ell \mathbb{E}\hat{\mathfrak{R}}_S(\tilde{\mathcal{H}}) \le C\mu_\ell \mathbb{E}\sqrt{\frac{\hat{R}_s}{n}} \le C\mu_\ell \sqrt{\frac{\mathbb{E}\hat{R}_s}{n}}.$$

To apply Corollary 3.5 of [20], we should solve the equation

$$r = C\mu_\ell \sqrt{\frac{r}{n}}, \tag{S.10}$$

and obtain $r^* = \frac{\mu_\ell^2 C^2}{n}$. Thus, for any $\eta > 0$, with probability at least $1 - e^{-\eta}$, there exists a constant $C' > 0$ that satisfies

$$P_n \ell(y, h) \le C'\left( \mathbb{E}\hat{R}_s + \frac{\mu_\ell^2 C^2}{n} + \frac{\eta}{n} \right) \le C'\left( R_s + \frac{\mu_\ell^2 C^2}{n} + \frac{\eta}{n} \right). \tag{S.11}$$

Note that, for the last inequality, because $\hat{R}_s \le P_n \ell(y, \langle \boldsymbol{\alpha}, \Phi_1 \rangle) + \lambda_{\boldsymbol{\alpha}} \|\boldsymbol{\alpha}\|^2$ for any $\boldsymbol{\alpha}$, taking the expectation of both sides yields $\mathbb{E}\hat{R}_s \le P\ell(y, \langle \boldsymbol{\alpha}, \Phi_1 \rangle) + \lambda_{\boldsymbol{\alpha}} \|\boldsymbol{\alpha}\|^2$, and this gives $\mathbb{E}\hat{R}_s \le \inf_{\boldsymbol{\alpha}}\{P\ell(y, \langle \boldsymbol{\alpha}, \Phi_1 \rangle) + \lambda_{\boldsymbol{\alpha}} \|\boldsymbol{\alpha}\|^2\} = R_s$. Consequently, applying Theorem 1 of [45], we have

$$P\ell(y, h(\boldsymbol{x})) \le P_n \ell(y, h(\boldsymbol{x})) + \tilde{O}\left( \left( \sqrt{\frac{R_s}{n}} + \frac{\mu_\ell C + \sqrt{\eta}}{n} \right) \left( \sqrt{L}C + \sqrt{L\eta} \right) + \frac{C^2 L + L\eta}{n} \right). \tag{S.12}$$

Here, we use $\hat{\mathfrak{R}}_S(\tilde{\mathcal{H}}) \le C\sqrt{\frac{\hat{R}_s}{n}} \le C\sqrt{\frac{L}{n}}$. $\qquad\square$

*Remark D.1.* As in [11], without the estimation of the parameters $\boldsymbol{\alpha}$ and $\boldsymbol{\beta}$, the right-hand side of Eq. (S.9) becomes $\frac{1}{\sqrt{n}}\left( c_1 + c_2 \sqrt{\hat{R}_s} \right)$ with some constant $c_1 > 0$ and $c_2 > 0$, and Eq. (S.10) becomes

$$r = \frac{1}{\sqrt{n}}(c_1 + c_2\sqrt{r}).$$

This yields the solution

$$r^* = \left( \frac{c_2}{2\sqrt{n}} + \sqrt{\left( \frac{c_2}{2\sqrt{n}} \right)^2 + \frac{c_1}{\sqrt{n}}} \right)^2 \le \frac{c_2^2}{n} + \frac{c_1}{\sqrt{n}},$$

where we use the inequality $\sqrt{x} + \sqrt{x+y} \le \sqrt{4x+2y}$. Thus, Eq. (S.11) becomes

$$P_n \ell(y, h) \le C' \left( R_s + \frac{c_2^2}{n} + \frac{c_1}{\sqrt{n}} + \frac{\eta}{n} \right).$$

Consequently, we have the following result:

$$P\ell(y, h(\boldsymbol{x})) \le P_n \ell(y, h(\boldsymbol{x}))$$
$$+ \tilde{O} \left( \left( \sqrt{\frac{R_s}{n}} + \frac{\sqrt{c_1}}{n^{3/4}} + \frac{c_2 + \sqrt{\eta}}{n} \right) \left( c_1 + c_2 \sqrt{L} + \sqrt{L\eta} \right) + \frac{(c_1 + c_2\sqrt{L})^2 + L\eta}{n} \right).$$

This means that even if $R_s = \tilde{O}(n^{-1})$, the resulting rate only improves to $\tilde{O}(n^{-3/4})$.

### D.3  Proof of Theorem 4.4

Recall that loss function $\ell(\cdot, \cdot)$ is assumed to be $\mu_\ell$-Lipschitz for the first argument. In addition, we impose the following assumption.

**Assumption D.2.** There exists a constant $B \ge 1$ such that for every $h \in \mathcal{H}$, $P(h - h^*) \le BP(\ell(y, h) - \ell(y, h^*))$.

Because we consider $\ell(y, y') = (y - y')^2$ in Theorem 4.4, Assumption D.2 holds as stated in [20].

First, we show the following corollary, which is a slight modification of Theorem 5.4 of [20].

**Corollary D.3.** *Let* $\hat{h}$ *be any element of* $\mathcal{H}$ *satisfying* $P_n \ell(y, \hat{h}) = \inf_{h \in \mathcal{H}} P_n \ell(y, h)$, *and let* $\hat{h}^{(m)}$ *be any element of* $\mathcal{H}^{(m)}$ *satisfying* $P_n \ell(y, \hat{h}^{(m)}) = \inf_{h \in \mathcal{H}^{(m)}} P_n \ell(y, h)$. *Define*

$$\hat{\psi}(r) = c_1 \hat{\mathfrak{R}}_S \{ h \in \mathcal{H} : \max_{m \in \{1,2\}} P_n (h^{(m)} - \hat{h}^{(m)})^2 \le c_3 r \} + \frac{c_2 \eta}{n},$$

*where* $c_1, c_2$ *and* $c_3$ *are constants depending only on* $B$ *and* $\mu_\ell$. *Then, for any* $\eta > 0$, *with probability at least* $1 - 5e^{-\eta}$,

$$P\ell(y, \hat{h}) - P\ell(y, h^*) \le \frac{705}{B} \hat{r}^* + \frac{(11\mu_\ell + 27B)\eta}{n},$$

*where* $\hat{r}^*$ *is the fixed point of* $\hat{\psi}$.

*Proof.* Define the function $\psi$ as

$$\psi(r) = \frac{c_1}{2} \mathfrak{R} \{ h \in \mathcal{H} : \mu_\ell^2 \max P(h^{(m)} - h^{(m)*})^2 \le r \} + \frac{(c_2 - c_1)\eta}{n}.$$

Because $\mathcal{H}, \mathcal{H}^{(1)}$ and $\mathcal{H}^{(2)}$ are all convex and thus star-shaped around each of its points, Lemma 3.4 of [20] implies that $\psi$ is a sub-root. Also, define the sub-root function $\psi_m$ as

$$\psi_m(r) = \frac{c_1^{(m)}}{2} \mathfrak{R} \{ h^{(m)} \in \mathcal{H}^{(m)} : \mu_\ell^2 P(h^{(m)} - h^{(m)*})^2 \le r \} + \frac{(c_2 - c_1)\eta}{n}.$$

Let $r_m^*$ be the fixed point of $\psi_m(r_m)$. Now, for $r_m \ge \psi_m(r_m)$, Corollary 5.3 of [20] and the condition on the loss function imply that, with probability at least $1 - e^{-\eta}$,

$$\mu_\ell^2 P(\hat{h}^{(m)} - h^{(m)*})^2 \le B\mu_\ell^2 P(\ell(y, \hat{h}^{(m)}) - \ell(y, \hat{h}^{(m)*})) \le 705\mu_\ell^2 r_m + \frac{(11\mu_\ell + 27B)B\mu_\ell^2 \eta}{n}.$$

Denote the right-hand side by $s_m$, and define $r = \max r_m$ and $s = \max s_m$. Because $s \ge s_m \ge r_m \ge r_m^*$, we obtain $s \ge \psi_m(s)$ according to Lemma 3.2 of [20], and thus,

$$s \ge 10\mu_\ell^2 \mathfrak{R} \{ h^{(m)} \in \mathcal{H}^{(m)} : \mu_\ell^2 P(h^{(m)} - h^{(m)*})^2 \le s \} + \frac{11\mu_\ell^2 \eta}{n}.$$

Therefore, applying Corollary 2.2 of [20] to the class $\mu_\ell \mathcal{H}^{(m)}$, it follows that with probability at least $1 - e^{-\eta}$,

$$\{h^{(m)} \in \mathcal{H}^{(m)} : \mu_\ell^2 P(h^{(m)} - h^{(m)*})^2 \le s\} \subseteq \{h^{(m)} \in \mathcal{H}^{(m)} : \mu_\ell^2 P_n(h^{(m)} - h^{(m)*})^2 \le 2s\}.$$

This implies that with probability at least $1 - 2e^{-\eta}$,

$$P_n(\hat{h}^{(m)} - h^{(m)*})^2 \le 2\left(705r + \frac{(11\mu_\ell + 27B)B\eta}{n}\right)$$

$$\le 2\left(705 + \frac{(11\mu_\ell + 27B)B}{n}\right)r,$$

where the second inequality follows from $r \ge \psi(r) \ge \frac{c_2\eta}{n}$. Define $2\left(705 + \frac{(11\mu_\ell + 27B)B}{n}\right) = c'$. According to the triangle inequality in $L_2(P_n)$, it holds that

$$P_n(h^{(m)} - \hat{h}^{(m)})^2 \le \left(\sqrt{P_n(h^{(m)} - h^{(m)*})^2} + \sqrt{P_n(h^{(m)*} - \hat{h}^{(m)})^2}\right)^2$$

$$\le \left(\sqrt{P_n(h^{(m)} - h^{(m)*})^2} + \sqrt{c'r}\right)^2.$$

Again, applying Corollary 2.2 of [20] to $\mu_\ell \mathcal{H}^{(m)}$ as before, but now for $r \ge \psi_m(r)$, it follows that with probability at least $1 - 4e^{-\eta}$,

$$\{h \in \mathcal{H} : \mu_\ell^2 \max P(h^{(m)} - h^{(m)*})^2 \le r\}$$

$$= \bigcap_{m=1}^2 \{h^{(m)} \in \mathcal{H}^{(m)} : \mu_\ell^2 P(h^{(m)} - h^{(m)*})^2 \le r\}$$

$$\subseteq \bigcap_{m=1}^2 \{h^{(m)} \in \mathcal{H}^{(m)} : \mu_\ell^2 P_n(h^{(m)} - h^{(m)*})^2 \le 2r\}$$

$$\subseteq \bigcap_{m=1}^2 \{h^{(m)} \in \mathcal{H}^{(m)} : \mu_\ell^2 P_n(h^{(m)} - \hat{h}^{(m)})^2 \le (\sqrt{2r} + \sqrt{c'r})^2\}$$

$$= \{h \in \mathcal{H} : \mu_\ell^2 \max P_n(h^{(m)} - \hat{h}^{(m)})^2 \le c_3 r\},$$

where $c_3 = (\sqrt{2} + \sqrt{c'})^2$. Combining this with Lemma A.4 of [20] leads to the following inequality: with probability at least $1 - 5e^{-x}$

$$\psi(r) = \frac{c_1}{2}\mathfrak{R}\{h \in \mathcal{H} : \mu_\ell^2 \max P(h^{(m)} - h^{(m)*})^2 \le r\} + \frac{(c_2 - c_1)\eta}{n}$$

$$\le c_1\hat{\mathfrak{R}}_S\{h \in \mathcal{H} : \mu_\ell^2 \max P(h^{(m)} - h^{(m)*})^2 \le r\} + \frac{c_2\eta}{n}$$

$$\le c_1\hat{\mathfrak{R}}_S\{h \in \mathcal{H} : \mu_\ell^2 \max P_n(h^{(m)} - \hat{h}^{(m)})^2 \le c_3 r\} + \frac{c_2\eta}{n}$$

$$= \hat{\psi}(r).$$

Letting $r = r^*$ and using Lemma 4.3 of [20], we obtain $r^* \le \hat{r}^*$, thus proving the statement. $\quad\square$

Under Assumption 4.2, we obtain the following excess risk bound for the proposed model class using Corollary D.3. The proof is based on [20].

**Theorem D.4.** *Let $\hat{h}$ be any element of $\mathcal{H}$ satisfying $P_n\ell(y, \hat{h}(\boldsymbol{x})) = \inf_{h \in \mathcal{H}} P_n\ell(y, h(\boldsymbol{x}))$. Suppose that Assumption 4.2 is satisfied, then there exists a constant $c$ depending only on $\mu_\ell$ such that for any $\eta > 0$, with probability at least $1 - 5e^{-\eta}$,*

$$P(y - \hat{h}(\boldsymbol{x}))^2 - P(y - h^*(\boldsymbol{x}))^2$$

$$\le c\left(\min_{0 \le \kappa_1, \kappa_2 \le n}\left\{\frac{\kappa_1 + \kappa_2}{n} + \left(\frac{1}{n}\sum_{j=\kappa_1+1}^n \hat{\lambda}_j^{(1)} + \sum_{j=\kappa_2+1}^n \hat{\lambda}_j^{(2)}\right)^{\frac{1}{2}}\right\} + \frac{\eta}{n}\right).$$

Theorem D.4 is a multiple-kernel version of Corollary 6.7 of [20], and a data-dependent version of Theorem 2 of [46] which considers the eigenvalues of the Hilbert-Schmidt operators on $\mathcal{H}$ and $\mathcal{H}^{(m)}$. Theorem D.4 concerns the eigenvalues of the Gram matrices $K^{(m)}$ computed from the data.

*Proof of Theorem D.4.* Define $R = \max_m \sup_{h \in \mathcal{H}^{(m)}} P_n(y - h(\boldsymbol{x}))^2$. For any $h \in \mathcal{H}^{(m)}$, we obtain

$$P_n(h^{(m)}(\boldsymbol{x}) - \hat{h}^{(m)}(\boldsymbol{x}))^2 \leq 2P_n(y - h^{(m)}(\boldsymbol{x}))^2 + 2P_n(y - \hat{h}^{(m)}(\boldsymbol{x}))^2 \leq 4 \sup_{h \in \mathcal{H}^{(m)}} P_n(y - h(\boldsymbol{x}))^2 \leq 4R.$$

From the symmetry of the $\sigma_i$ and the fact that $\mathcal{H}^{(m)}$ is convex and symmetric, we obtain the following:

$$\hat{\mathfrak{R}}_S\{h \in \mathcal{H} : \max P_n(h^{(m)} - \hat{h}^{(m)})^2 \leq 4R\}$$

$$= \mathbb{E}_\sigma \sup_{\substack{h^{(m)} \in \mathcal{H}^{(m)} \\ P_n(h^{(m)} - \hat{h}^{(m)})^2 \leq 4R}} \frac{1}{n} \sum_{i=1}^n \sigma_i \sum_{m=1}^2 h^{(m)}(\boldsymbol{x}_i)$$

$$= \mathbb{E}_\sigma \sup_{\substack{h^{(m)} \in \mathcal{H}^{(m)} \\ P_n(h^{(m)} - \hat{h}^{(m)})^2 \leq 4R}} \frac{1}{n} \sum_{i=1}^n \sigma_i \sum_{m=1}^2 (h^{(m)}(\boldsymbol{x}_i) - \hat{h}^{(m)}(\boldsymbol{x}_i))$$

$$\leq \mathbb{E}_\sigma \sup_{\substack{h^{(m)}, g^{(m)} \in \mathcal{H}^{(m)} \\ P_n(h^{(m)} - g^{(m)})^2 \leq 4R}} \frac{1}{n} \sum_{i=1}^n \sigma_i \sum_{m=1}^2 (h^{(m)}(\boldsymbol{x}_i) - g^{(m)}(\boldsymbol{x}_i))$$

$$= 2\mathbb{E}_\sigma \sup_{\substack{h^{(m)} \in \mathcal{H}^{(m)} \\ P_n h^{(m)2} \leq R}} \frac{1}{n} \sum_{i=1}^n \sigma_i \sum_{m=1}^2 h^{(m)}(\boldsymbol{x}_i)$$

$$\leq 2 \sum_{m=1}^2 \mathbb{E}_\sigma \sup_{\substack{h^{(m)} \in \mathcal{H}^{(m)} \\ P_n h^{(m)2} \leq R}} \frac{1}{n} \sum_{i=1}^n \sigma_i h^{(m)}(\boldsymbol{x}_i)$$

$$\leq 2 \sum_{m=1}^2 \left\{ \frac{2}{n} \sum_{j=1}^n \min\{R, \hat{\lambda}_j^{(m)}\} \right\}^{\frac{1}{2}}$$

$$\leq \left\{ \frac{16}{n} \sum_{m=1}^2 \sum_{j=1}^n \min\{R, \hat{\lambda}_j^{(m)}\} \right\}^{\frac{1}{2}}.$$

The second inequality comes from the subadditivity of supremum and the third inequality follows from Theorem 6.6 of [20]. To obtain the last inequality, we use $\sqrt{x} + \sqrt{y} \leq \sqrt{2(x + y)}$. Thus, we have

$$2c_1 \hat{\mathfrak{R}}_S\{h \in \mathcal{H} : \max P_n(h^{(m)} - \hat{h}^{(m)})^2 \leq 4R\} + \frac{(c_2 + 2)\eta}{n}$$

$$\leq 4c_1 \left\{ \frac{16}{n} \sum_{m=1}^2 \sum_{j=1}^n \min\left\{R, \hat{\lambda}_j^{(m)}\right\} \right\}^{\frac{1}{2}} + \frac{(c_2 + 2)\eta}{n},$$

for some constants $c_1$ and $c_2$. To apply Corollary D.3, we should solve the following inequality for $r$

$$r \leq 4c_1 \left\{ \frac{16}{n} \sum_{m=1}^2 \sum_{j=1}^n \min\left\{r, \hat{\lambda}_j^{(m)}\right\} \right\}^{\frac{1}{2}}.$$

For any integers $\kappa_m \in [0, n]$, the right-hand side is bounded as

$$4c_1 \left\{ \frac{16}{n} \sum_{m=1}^{2} \sum_{j=1}^{n} \min\left\{ r, \hat{\lambda}_j^{(m)} \right\} \right\}^{\frac{1}{2}} \leq 4c_1 \left\{ \frac{16}{n} \sum_{m=1}^{2} \left( \sum_{j=1}^{\kappa_m} r + \sum_{j=\kappa_m+1}^{n} \hat{\lambda}_j^{(m)} \right) \right\}^{\frac{1}{2}}$$

$$= \left\{ \left( \frac{256c_1^2}{n} \sum_{m=1}^{2} \kappa_m \right) r + \frac{256c_1^2}{n} \sum_{m=1}^{2} \sum_{j=\kappa_m+1}^{n} \hat{\lambda}_j^{(m)} \right\}^{\frac{1}{2}},$$

and we obtain the solution $r^*$ as

$$r^* \leq \frac{128c_1^2}{n} \sum_{m=1}^{2} \kappa_m + \left( \left\{ \frac{128c_1^2}{n} \sum_{m=1}^{2} \kappa_m \right\}^2 + \frac{256c_1^2}{n} \sum_{m=1}^{2} \sum_{j=\kappa_m+1}^{n} \hat{\lambda}_j^{(m)} \right)^{\frac{1}{2}}$$

$$\leq \frac{256c_1^2}{n} \sum_{m=1}^{2} \kappa_m + \left( \frac{256c_1^2}{n} \sum_{m=1}^{2} \sum_{j=\kappa_m+1}^{n} \hat{\lambda}_j^{(m)} \right)^{\frac{1}{2}}.$$

Optimizing the right-hand side with respect to $\kappa_1$ and $\kappa_2$, we obtain the solution as

$$r^* \leq \min_{0 \leq \kappa_1, \kappa_2 \leq n} \left\{ \frac{256c_1^2}{n} \sum_{m=1}^{2} \kappa_m + \left( \frac{256c_1^2}{n} \sum_{m=1}^{2} \sum_{j=\kappa_m+1}^{n} \hat{\lambda}_j^{(m)} \right)^{\frac{1}{2}} \right\}.$$

Furthermore, according to Corollary D.3, there exists a constant $c$ such that with probability at least $1 - 5e^{-\eta}$,

$$P(y - \hat{h}(x))^2 - P(y - h^*(x))^2$$

$$\leq c \left( \min_{0 \leq \kappa_1, \kappa_2 \leq n} \left\{ \frac{1}{n} \sum_{m=1}^{2} \kappa_m + \left( \frac{1}{n} \sum_{m=1}^{2} \sum_{j=\kappa_m+1}^{n} \hat{\lambda}_j^{(m)} \right)^{\frac{1}{2}} \right\} + \frac{\eta}{n} \right).$$

$\square$

With Theorem D.4 and Assumption 4.3, we prove Theorem D.4 as follows.

*Proof of Theorem 4.4.* Using the inequality $\sqrt{x + y} \leq \sqrt{x} + \sqrt{y}$ for $x \geq 0, y \geq 0$, we have

$$P(y - \hat{h}(\boldsymbol{x}))^2 - P(y - h^*(\boldsymbol{x}))^2$$

$$= O\left( \min_{0 \leq \kappa_1, \kappa_2 \leq n} \left\{ \frac{\kappa_1 + \kappa_2}{n} + \left( \frac{1}{n} \sum_{j=\kappa_1+1}^{n} \hat{\lambda}_j^{(1)} + \frac{1}{n} \sum_{j=\kappa_2+1}^{n} \hat{\lambda}_j^{(2)} \right)^{\frac{1}{2}} \right\} + \frac{\eta}{n} \right)$$

$$\leq O\left( \min_{0 \leq \kappa_1, \kappa_2 \leq n} \left\{ \frac{\kappa_1 + \kappa_2}{n} + \left( \frac{1}{n} \sum_{j=\kappa_1+1}^{n} j^{-\frac{1}{s_1}} + \frac{1}{n} \sum_{j=\kappa_2+1}^{n} j^{-\frac{1}{s_2}} \right)^{\frac{1}{2}} \right\} + \frac{\eta}{n} \right)$$

$$\leq O\left( \min_{0 \leq \kappa_1, \kappa_2 \leq n} \left\{ \frac{\kappa_1 + \kappa_2}{n} + \left( \frac{1}{n} \sum_{j=\kappa_1+1}^{n} j^{-\frac{1}{s_1}} \right)^{\frac{1}{2}} + \left( \frac{1}{n} \sum_{j=\kappa_2+1}^{n} j^{-\frac{1}{s_2}} \right)^{\frac{1}{2}} \right\} + \frac{\eta}{n} \right).$$

Because it holds that

$$\sum_{j=\kappa_m+1}^{n} j^{-\frac{1}{s_m}} < \int_{\kappa_m}^{\infty} x^{-\frac{1}{s_m}} dx < \left[ \frac{1}{1 - \frac{1}{s_m}} x^{1-\frac{1}{s_m}} \right]_{\kappa_m}^{\infty} = \frac{s_m}{1 - s_m} \kappa_m^{1-\frac{1}{s_m}},$$

for $m = 1, 2$, we should solve the following minimization problem:

$$\min_{0 \le \kappa_1, \kappa_2 \le n} \left\{ \frac{\kappa_1 + \kappa_2}{n} + \left( \frac{1}{n} \frac{s_1}{1 - s_1} \kappa_1^{1 - \frac{1}{s_1}} \right)^{\frac{1}{2}} + \left( \frac{1}{n} \frac{s_1}{1 - s_1} \kappa_2^{1 - \frac{1}{s_1}} \right)^{\frac{1}{2}} \right\} \equiv g(\kappa).$$

Taking the derivative, we have

$$\frac{\partial g(\kappa)}{\partial \kappa_1} = \frac{1}{n} + \frac{1}{2} \left( \frac{1}{n} \frac{s_1}{1 - s_1} \kappa_1^{1 - \frac{1}{s_1}} \right)^{-\frac{1}{2}} \left( -\frac{\kappa_1^{-\frac{1}{s_1}}}{n} \right).$$

Setting this to zero, we find the optimal $\kappa_1$ as

$$\kappa_1 = \left( \frac{s_1}{1 - s_1} \frac{4}{n} \right)^{\frac{s_1}{1 + s_1}}.$$

Similarly, we have

$$\kappa_2 = \left( \frac{s_2}{1 - s_2} \frac{4}{n} \right)^{\frac{s_2}{1 + s_2}},$$

and

$$P(y - \hat{h}(\boldsymbol{x}))^2 - P(y - h^*(\boldsymbol{x}))^2$$

$$\le O\left( \frac{1}{n} \left( \frac{s_1}{1 - s_1} \frac{4}{n} \right)^{\frac{s_1}{1 + s_1}} + \frac{1}{n} \left( \frac{s_2}{1 - s_2} \frac{4}{n} \right)^{\frac{s_2}{1 + s_2}} \right.$$

$$\left. + 2^{\frac{1 - s_1}{1 + s_1}} \left( \frac{s_1}{1 - s_1} \frac{1}{n} \right)^{\frac{1}{1 + s_1}} + 2^{\frac{1 - s_2}{1 + s_2}} \left( \frac{s_2}{1 - s_2} \frac{1}{n} \right)^{\frac{1}{1 + s_2}} + \frac{\eta}{n} \right)$$

$$= O\left( n^{-\frac{1}{1 + s_1}} + n^{-\frac{1}{1 + s_2}} \right)$$

$$= O\left( n^{-\frac{1}{1 + \max\{s_1, s_2\}}} \right).$$

$\square$

