# OpenReview forum: "Transfer Learning with Affine Model Transformation"
_NeurIPS.cc/2023/Conference — NeurIPS 2023 poster_

### Official Review · Reviewer_t1vu · 2023-07-06

**Soundness:** 3 good
**Presentation:** 3 good
**Contribution:** 3 good
**Rating:** 6
**Confidence:** 2

**Summary:**

The paper provides a comprehensive investigation of transfer learning and proposes an affine model transfer framework. The authors summarize various transfer learning approaches under the class of affine model transformation and establish theoretical properties such as generalization error and excess risk.
In the experimental evaluation, the proposed methods demonstrate comparable performance to other baseline methods on tasks such as robot arm trajectory prediction and scientific document score prediction.


**Strengths:**

The paper studies an important topic: the theoretical properties of transfer learning.

The proposed theoretical framework for affine model transfer and the derived generalization bound provide valuable insights into transfer learning, domain generalization, and model adaptation.

Overall, the paper is well-organized and effectively conveys the intricacies of the proposed theoretical framework.


**Weaknesses:**

My main concerns revolve around the experimental evaluation conducted in the paper. The evaluation is narrower in scope compared to most transfer learning studies. It would be beneficial to validate the proposed method on standard transfer learning benchmarks, such as the MIT Indoors dataset, Caltech 256 dataset, or other commonly used regression datasets. This kind of evaluation can be found in references [1,2].
Furthermore, in the experimental evaluation, the observed improvement is limited. Table 1, specifically in the Torque 7 prediction, shows that the proposed method performs worse than Fine-tuning in most cases.

While Theorem 2.4 provides a nice property of transformation functions, the paper lacks a discussion on the specific properties of $g_1, g_2$, and $g_3$ functions. It would be helpful to provide suggestions or insights into the types of $g_1, g_2$, and  $g_3$ functions.

[1] L. Xuhong, et al, “Explicit inductive bias for transfer learning with convolutional networks,” ICML, 2018.
[2] S. Myung, et al, “PAC-Net: A model pruning approach to inductive transfer learning,” ICML, 2022.


**Questions:**

I would like to know that why were the experiments not conducted on commonly adopted transfer learning datasets, such as MIT Indoors, Caltech 256, and CelebA?

Considering that the proposed method performs worse than Fine-tuning in the case of Torque 7 prediction according to table 1, it is important to clarify the advantages of the proposed method over Fine-tuning. What unique benefits or improvements does the proposed method offer in comparison?

Are there any suggestions provided regarding the design of $g_1, g_2$, and  $g_3$ functions, or any specific neural network architectures? Providing guidance on the selection and design of these components would greatly assist readers in implementing the proposed method effectively.


**Limitations:**

See [Weaknesses] for limitations.

---

> ### Author Rebuttal · Authors · 2023-08-09
>
> We express our gratitude to the reviewer for providing us with valuable feedback. Below is our point-by-point response to each question.
>
> > **[Q1]** I would like to know that why were the experiments not conducted on commonly adopted transfer learning datasets, such as MIT Indoors, Caltech 256, and CelebA?
>
> The MIT Indoors, Caltech 256, and CelebA datasets are primarily designed for classification and object recognition tasks, not for regression tasks which are the main focus of this study. Transfer learning for regression has been less studied than for classification tasks. Hence, common benchmark datasets have not been established. For example, Li et al. (2018) limited their experiments to object recognition and scene classification. Myung et al. (2022) conducted experiments on regression tasks, but used only simplistic toy examples rather than real-world datasets.
> Therefore, we conducted two real-world experiments with our original datasets in the field of physical chemistry in Supplementary Materials. We aim to provide these two datasets to the machine learning community as new public datasets for transfer learning regression.
>
> > **[Q2]** Considering that the proposed method performs worse than Fine-tuning in the case of Torque 7 prediction according to table 1, it is important to clarify the advantages of the proposed method over Fine-tuning. What unique benefits or improvements does the proposed method offer in comparison?
>
> The strength of our method compared to fine-tuning is that it does not degrade its performance in cases where cross-domain relationships are weak. While the experiment in Section 5.1 revealed that fine-tuning outperformed our method in cases of Torque 7, the performance of fine-tuning was significantly degraded as the source-target relationship became weaker, as seen in Torque 1 case. In contrast, our method was able to avoid negative transfer even for such cases. This characteristic is particularly beneficial because, in many cases, the degree of relatedness between the domains is not known in advance.
> Another distinctive feature of our method is its theoretical foundation of optimality for the squared loss case. This aspect would encourage further development of transfer learning for regression.
> In the revised manuscript, we will provide additional discussion comparing our method to weight-based transfer learning, as replied above.
>
> > **[Q3]** Are there any suggestions provided regarding the design of $g_1,g_2$, and $g_3$ functions, or any specific neural network architectures? Providing guidance on the selection and design of these components would greatly assist readers in implementing the proposed method effectively.
>
> One of our key principles for the design of $g_1,g_2$, and $g_3$ is interpretability. In our model, $g_1$ and $g_2$ primarily facilitate knowledge transfer, while the estimated $g_3$ is used to gain insight on domain-specific factors. For instance, in order to infer cross-domain differences, we could design $g_1$ and $g_2$ by the conventional neural feature extraction, and a simple, highly interpretable model such as a linear model could be used for $g_3$. Thus, observing the estimated regression coefficients in $g_3$, one can statistically infer which features of x are related to inter-domain differences. This advantage of the proposed method is demonstrated in Section 5.2 and in Section B.3.
> In the revised manuscript, we will provide such additional guidance on the design of each component of the model.
>
> [Reference]
> X. Li, et al. “Explicit inductive bias for transfer learning with convolutional networks,” ICML, 2018.
> S. Myung, et al. “PAC-Net: A model pruning approach to inductive transfer learning,” ICML, 2022.

---

> > ### Comment · Reviewer_t1vu · 2023-08-14
> > **Appreciation for the Author's Response**
> >
> > I extend my gratitude for the author's comprehensive response, which has indeed addressed some of my concerns. However, the observed inferior performance of the proposed method in certain settings, as exemplified by the Torque 7 prediction scenario in the paper, raises a pertinent question. When faced with a specific dataset, what criteria should guide the decision between employing the proposed method versus other available approaches? There's a genuine concern that an inadvertent application of the proposed approach might result in outcomes worse than simple fine-tuning.
> > Hence, I kindly request the authors to offer specific guidelines or criteria for determining when the proposed approach is most suitable.

---

> > > ### Author Response · Authors · 2023-08-15
> > > **Thanks for your response.**
> > >
> > > There are no definitive criteria to determine whether our method or fine-tuning should be used. However, our approach is well-suited when feature-based methods are adopted.
> > > Weight-based methods can sometimes be unsuitable, particularly when transferring knowledge from huge models, such as LLMs. In these scenarios, fine-tuning all parameters is not feasible. Instead of using parameters, source information may be provided as additional features, as seen in Section B.3. In such situations, feature-based transfer learning is preferred. Our approach often outperforms other feature-based methods, as demonstrated in Section 5.1.
> > > In our revised version, we will include these guidelines to assist readers in assessing the applicability of our proposed method.

---

> > > > ### Comment · Reviewer_t1vu · 2023-08-16
> > > > **Pots-rebuttal review**
> > > >
> > > > Thank you for your thorough response to my queries.
> > > > After carefully reviewing the paper and taking into account the insights provided by both the authors and fellow reviewers, I am convinced of the practical value of this paper within the domain of Transfer Learning. As a result, I am inclined to revise my initial rating from 5 (Borderline Accept) to 6 (Weak Accept).

---

> > > > > ### Author Response · Authors · 2023-08-17
> > > > >
> > > > > We are pleased to hear that our response has addressed your concerns. Thank you again for taking the time to respond.

---

### Official Review · Reviewer_3FfK · 2023-07-06

**Soundness:** 3 good
**Presentation:** 3 good
**Contribution:** 2 fair
**Rating:** 6
**Confidence:** 3

**Summary:**

The authors consider the transfer learning setting by affine models adapting source features to the target domain by separately estimating cross-domain shift and domain-specific factors. They motivate the choice of such models deriving the affine form as an optimal class based on the squared loss for the target task. The authors show that the considered affine model encompasses several existing transfer learning methods, including neural network based transfer learning. The authors provide an estimation algorithm when modeled by kernel methods. The authors provide a generalization bound and an excess risk bound for their method. The bound show that when the source feature is strongly related to the target domain, their transfer learning approach results to be advantageous in comparison to the vanilla learning approach. Such an advantage is confirmed also by numerical experiments, in which the proposed method outperforms widely applied transfer learning methods lacking of theoretical foundation, such as re-using features from pre-trained models. More specifically, they consider two experimental settings: (i) the prediction of feed-forward torque at seven joints of the robot arm, and (ii) the prediction of review scores and decisions of scientific papers. Additionally, two case studies in materials science are presented in the appendix.

**Strengths:**

The authors consider an interesting problem in a rigorous and formal way.

The authors provide both theoretical and experimental validation for their method.

The derivation/motivation of the affine form as an optimal class based on the squared loss for the target task is interesting.

The affine model transfer is model-agnostic and it can be easily combined with pre-existent models. In addition, since domain specific factors are involved in incorporating the source features, the affine transfer can also deal with domain common and unique factors simultaneously and separately.

Overall, the paper is written in a clear way.

**Weaknesses:**

Considering only the squared loss is quite limiting. The authors comment about this, but it would be nice to understand if there is possibility to extend their framework to more general loss functions. For instance, what about the optimality of the class of affine functions with different loss functions? The authors say they can also derive the same optimal function by minimizing the upper bound of the estimation error in the TL procedure, as discussed in the appendix. I guess this could be related to my question.

Some technical aspects could be better explained and intuitively motivated in my opinion, see for instance my comment below.

**Questions:**

The algorithm the authors propose applies a coordinate descent minimization procedure on the objective function F. Is F jointly convex? Is the algorithm globally convergent? Usually, the study of global convergence for coordinate descent algorithms is problematic.

The authors require the Lipschitzness of the loss function, but the squared loss is Lipschitz if and only if the labels are bounded and this is quite restrictive.

From Thm. 4.1 it seems the authors get an improvement w.r.t. n by transfer learning. This seems to be strange, since, usually, the advantage observed in previous work by transfer learning is by constant factors. Could you please comment about this with reference to the papers 'Stability and hypothesis transfer learning' and 'Fast rates by transferring from auxiliary hypotheses'?

**Limitations:**

I do not see any potential negative societal impact related to this work.

---

> ### Author Rebuttal · Authors · 2023-08-09
>
> Thank you very much for the valuable comments. Below we respond to each concern / question raised in your review.
>
> > **[W1]** Considering only the squared loss is quite limiting. The authors comment about this, but it would be nice to understand if there is possibility to extend their framework to more general loss functions. For instance, what about the optimality of the class of affine functions with different loss functions? The authors say they can also derive the same optimal function by minimizing the upper bound of the estimation error in the TL procedure, as discussed in the appendix. I guess this could be related to my question.
>
> In Section A.1, we discussed the case of general convex loss functions; it was found that the optimality of the affine function does not hold for other losses.
> In addition, further assumptions would be needed if the current framework is extended to include other losses based on the upper bound on the estimation error, as you suggest.
>
> > **[Q1]** The algorithm the authors propose applies a coordinate descent minimization procedure on the objective function F. Is F jointly convex? Is the algorithm globally convergent? Usually, the study of global convergence for coordinate descent algorithms is problematic.
>
> In Algorithm 1, $F$ is not jointly convex in general. However, the key consideration for running Algorithm 1 is that when two parameters are fixed, $F$ is convex with respect to the remaining parameter. This condition holds for kernel methods and generalized linear models.
> According to Zhou et al. (2013), when each sub-minimization problem is convex, Algorithm 1 is guaranteed to converge to a stationary point. Furthermore, Zhou et al. (2013) showed the consistency and asymptotic normality to hold for the alternating minimization algorithm.
> This discussion will be included in the revised version of the paper.
>
> > **[Q2]** The authors require the Lipschitzness of the loss function, but the squared loss is Lipschitz if and only if the labels are bounded and this is quite restrictive.
>
> Indeed, the assumption that the labels are bounded does not always hold. However, without this assumption, theoretical analysis becomes more challenging. For example, Mohri et al. (2018) states "The analysis of unbounded regression problems is technically more elaborate and typically requires some other types of assumptions." We believe that the labels are often bounded in many real-world problems.
>
> > **[Q3]** From Thm. 4.1 it seems the authors get an improvement w.r.t. n by transfer learning. This seems to be strange, since, usually, the advantage observed in previous work by transfer learning is by constant factors. Could you please comment about this with reference to the papers 'Stability and hypothesis transfer learning' and 'Fast rates by transferring from auxiliary hypotheses'?
>
> Theorem 4.1 in our study corresponds to Theorem 8 in Kuzborskij and Orabona (2017). The structure of the bounds derived in the two studies is identical. Both results indicate that the rate improves with respect to $n$ when the source features have high descriptive power for the target domain.
> An example of a bound with a constant factor can be found in Theorem 3 of Ben-David et al. (2010), where a constant factor appears as the $\mathcal{H}$-divergence. In that study, the authors considered a loss function of $(1-\alpha) \cdot (\text{error on the target task})+\alpha \cdot (\text{error on the source task})$. Note that our setting corresponds to $\alpha=0$. It is then found that the constant factor vanishes in the bound derived in Ben-David et al. (2010).
>
> [Reference]
> H. Zhou, et al. "Tensor regression with applications in neuroimaging data analysis,” Journal of the American Statistical Association, 2013.
> M. Mohri, et al. Foundations of machine learning. MIT press, 2018.
> I. Kuzborskij and F. Orabona, “Fast rates by transferring from auxiliary hypotheses,” Machine Learning, 2017.
> S. Ben-David, et al. "A theory of learning from different domains." Machine Learning, 2010.

---

> > ### Comment · Reviewer_3FfK · 2023-08-17
> > **Response to the authors**
> >
> > I thank the authors for their response. They answered my questions.

---

### Official Review · Reviewer_XSgT · 2023-07-07

**Soundness:** 3 good
**Presentation:** 3 good
**Contribution:** 2 fair
**Rating:** 4
**Confidence:** 4

**Summary:**

This paper presents a simple yet effective approach to transfer learning in regression problems by introducing an affine transform model. The proposed model incorporates the source feature function and the product of the source feature function and the function associated with the target input augmentation. Through the utilization of a least square loss function and a transformation function, this paper establishes the optimality of the affine model as the architecture of choice for addressing transfer learning challenges. Furthermore, the paper extends upon previous research by deriving generalization bounds and excess risk bounds in the proposed framework.

**Strengths:**

1. This paper is impeccably written and exceptionally comprehensible.

2. The proposed method is rigorously derived and its performance is systematically evaluated from a theoretical standpoint.

3. The proposed algorithm can incorporate and generalize previously proposed algorithms while showing better performance.

**Weaknesses:**

1. The existence of inverse functions appears to be of significance. However, it can be considered a potential limitation as it may not always be feasible to obtain inverse functions in practice.

2. I would suggest that the authors conduct evaluations of the proposed method using widely recognized benchmark datasets. It is particularly crucial to assess the algorithm's performance based on the proximity or disparity between the source and target, in order to ascertain its effectiveness under varying conditions.

**Questions:**

1. Would it not be an overstatement to claim that the proposed algorithm encompasses all existing algorithms? This assertion arises from the fact that weight-based algorithms (such as L2-SP and PAC-NET) are not incorporated into the proposed algorithm.

2. Could we possibly expand the experimentation to include a broader range of experimental datasets?

**Limitations:**

This reviewer thinks that the algorithm proposed in this paper has limitations that can only be applied to regression problems.

---

> ### Author Rebuttal · Authors · 2023-08-09
>
> Thank you for your time in writing in-depth feedback. Please find our response to your concerns below.
>
> > **[W1]** The existence of inverse functions appears to be of significance. However, it can be considered a potential limitation as it may not always be feasible to obtain inverse functions in practice.
>
> There may be a misunderstanding about Assumption 2.2. It assumes the invertibility of the function $\psi_{f_s} ∶ \mathbb{R} \to \mathbb{R}$ for a fixed $f_s (x)$. In other words, it assumes a one-to-one correspondence between the predictive value $\hat{f}_t (x)$ and the output of the intermediate model $\hat{g}(x)$, rather than between $x$ and $f_t$, or the source representation and target representation. If this assumption does not hold, then multiple values of $\hat{g}$ correspond to the same predicted value $\hat{f}_t$. Such modeling is unnatural. For example, if we set $\psi = g(x)^2+f_s (x)$, Assumption 2.2 does not hold. In this case, we get the same predicted value even if the sign of $\hat{g}(x)$ is reversed.
> In the revised manuscript, we will add the above explanation to clarify that Assumption 2.2 is never restrictive.
>
> > **[W2, Q2]** I would suggest that the authors conduct evaluations of the proposed method using widely recognized benchmark datasets. It is particularly crucial to assess the algorithm's performance based on the proximity or disparity between the source and target, in order to ascertain its effectiveness under varying conditions.
>
> Transfer learning for regression has been less studied than for classification tasks. Hence, commonly used benchmark datasets have not been established. Therefore, we conducted two real-world experiments with our original datasets in the field of physical chemistry in Supplementary Material. We aim to provide these two datasets to the machine learning community as new public datasets for transfer learning regression. It should also be noted that in one of the experiments, we investigated the effect of the proximity between the source features and target domain on the learning behavior of the affine model transfer.
> Due to the time constraints of this rebuttal, additional experiments can not be conducted. We will take your suggestion into consideration for future work.
>
> > **[Q1]** Would it not be an overstatement to claim that the proposed algorithm encompasses all existing algorithms? This assertion arises from the fact that weight-based algorithms (such as L2-SP and PAC-NET) are not incorporated into the proposed algorithm.
>
> As you pointed out, our method does not encompass weight-based transfer learning such as fine-tuning. Several misleading phrases, such as "including neural network-based TL" in line 53, will be corrected (e.g., "including the neural feature extraction").

---

> > ### Comment · Reviewer_XSgT · 2023-08-15
> > **Thank you for the author's response.**
> >
> > Thank you for the author's response.
> >
> > I've fully reviewed the author's response, but I would like to keep my score.
> >
> > The reason is that the contribution of this paper is limited; i) it seems to be applicable only to regression problem and ii) the assessed problems are not a well-known benchmark problem.
> >
> > I think it is essential to verify the proposed algorithm in well-known problems.

---

### Official Review · Reviewer_tCYn · 2023-07-27

**Soundness:** 3 good
**Presentation:** 3 good
**Contribution:** 3 good
**Rating:** 7
**Confidence:** 2

**Summary:**

This paper focuses on the theoretical perspective on the affine model transfer learning. Specifically, this paper presents the optimal class of models on transfer learning with l2 loss takes the form of an affine decoupling. The decoupling consists of three functions, \$g_1 + g_2g_3\$, where \$g_1, g_2\$ encodes the domain gaps between source and target domains, and \$g_3\$ encodes the domain-specific information. Further, for each of the functions, this paper proposes a kernel method to approximate. Additionally, the generalization and risk bounds are presented.

**___________________________________________________ Post Rebuttal __________________________________________________**


The reviewer has read the rebuttal.

In the rebuttal, parts of the theorems are restricted to \$ \mathbb{R} \$. This may degrade the contribution of the paper since we are handling the high-dimension data for most of the cases.

 The reviewer will maintain the score of 7 since there appear to be no technical flaws to the reviewer, and the reviewer gained some insights from the paper. However, the reviewer won't champion the paper, since I am not familiar to the literature.

**Strengths:**

1. This paper presents universal results to the transfer learning on linear regression, which looks interesting to the reviewer. Specifically, it is shown that the optimal solution under l2 loss falls into a certain class of models. And the analysis is not limited to machine learning models.
2. This paper also presents the generalization and risk bounds, making the analysis more solid.
2. Empirically, this paper experiments on two datasets with extensive ablation and baselines, making the paper complete and thorough.

**Weaknesses:**

1. Since the reviewer is not from the area, it is hard for the reviewer to judge the novelty and significance of the work.

**Questions:**

1. All the discussions are limited to the \$\mathbb{R}\$. It remains elusive whether the analysis can be extended to higher dimensional spaces naturally.
2. What is d in table 1, is it the source domain feature dimension?

**Limitations:**

While the paper's analysis is primarily limited to the regression task, it provides valuable insights with a commendable depth of exploration, despite limiting its general applicability. The complexity of a broader analysis would undoubtedly be a challenging undertaking, but the paper sets a strong foundation for future exploration.

---

> ### Author Rebuttal · Authors · 2023-08-09
>
> We are delighted to receive a positive evaluation from the reviewer. Below is our point-by-point response to each question.
>
> > **[Q1]** All the discussions are limited to the $\mathbb{R}$. It remains elusive whether the analysis can be extended to higher dimensional spaces naturally.
>
> When $y$ and $f_t$ are multi-dimensional, the claims in Theorem 2.4 would hold under some modifications. However, it is not obvious whether the theoretical analyses of Section 4 can be straightforwardly extended to the multi-dimensional scenario.
>
> > **[Q2]** What is d in table 1, is it the source domain feature dimension?
>
> $d$ is the dimension of the original input $x$. We will add the definition of $d$ in the caption of Table 1.

---

### Official Review · Reviewer_DK8u · 2023-07-28

**Soundness:** 3 good
**Presentation:** 2 fair
**Contribution:** 3 good
**Rating:** 5
**Confidence:** 1

**Summary:**

The authors focus on the hypothesis transfer learning (HTL) problem. They show that under some hypothesis the data and model transformation is given by an affine transformation. They consider an associated kernel regression problem to identify the transformation and then characterise the generalisation bound which reflects the importance of the target-source relation. Finally, the method is benchmarked in 2 settings.

**Strengths:**

* The proposed method allows for the theoretical characterisation of generalisation and excess risk bounds.
* The method is versatile and can be used in very different settings.

**Weaknesses:**

* The model is not clear (sec.2.1). Maybe adding a scheme or introducing the linear models as a first example before moving to the general version would help. Given this first setting, the rest flows nicely.
* I find assumption 2.2 a strong assumption. The authors comment saying that this is common in HTLs. However, it implies no information compression which is hard to believe in many practical contexts.
* In tables 1 and 2, the improvements are only marginally better than the other methods even if significant according to the authors' test.

Minor comments:
* assumption 2.2 (Invetribility) is instead (Invertibility)
* fig.1 is misplaced.
* proposition A.1 refer to the proof of Th.2.4 in the main paper, but this is instead in the SM sec.D.

**Questions:**

* In the formulation of sec.2.1, it seems that $f_t(x)$ and $f_s(x)$ have the same dimension of $y$, i.e. they both belong to $\mathbb R$. However, in the examples below $y$ has the same dimension as the scalar product $\langle \theta, f_s(x) \rangle$. Can you clarify?

**Limitations:**

I find assumption 2.2 a limitation. The authors seemed not concerned by this and pointed out that this is standard in the field. However, the intuition is that if the target representation is not included (or equal) to the source representation we shouldn't be able to invert the function.

---

> ### Author Rebuttal · Authors · 2023-08-09
>
> We would like to express our gratitude to the reviewer for providing insightful comments. Below is our point-by-point response to each of the comments and questions.
>
> > **[W1]** The model is not clear (sec.2.1). Maybe adding a scheme or introducing the linear models as a first example before moving to the general version would help. Given this first setting, the rest flows nicely.
>
> To make our setups easier to understand, we will add the following sentences to the beginning of Section 2.1 in the revised version:
>
> In this paper, we focus on transfer learning with variable transformations as proposed in Du et al. (2017). For an illustration of the concept, consider the case where there exists a relationship between the true functions $f_s^* (x) \in \mathbb{R}$ and $f_t^* (x) \in \mathbb{R}$ such that $f_t^* (x)=f_s^* (x)+x^\top \theta^*, x \in \mathbb{R}^d$ with an unknown parameter $\theta^* \in \mathbb{R}^d$. If $f_t^*$ is non-smooth, a large number of training samples is needed to learn $f_t^*$ directly. However, since the difference $f_t^*-f_s^*$ is a linear function with respect to the unknown $\theta^*$, it can be learned with fewer samples if prior information about $f_s^*$ is available. For example, a target model can be obtained by adding $f_s$ to the model $g$ trained for the intermediate variable $z=y-f_s (x)$.
>
> > **[W2]** I find assumption 2.2 a strong assumption. The authors comment saying that this is common in HTLs. However, it implies no information compression which is hard to believe in many practical contexts.
>
> There may be a misunderstanding about Assumption 2.2. It assumes the invertibility of the function $\psi_{f_s} ∶ \mathbb{R} \to \mathbb{R}$ for a fixed $f_s (x)$. In other words, it assumes a one-to-one correspondence between the predictive value $\hat{f}_t (x)$ and the output of the intermediate model $\hat{g}(x)$, rather than between $x$ and $f_t$, or the source representation and target representation. If this assumption does not hold, then multiple values of $\hat{g}$ correspond to the same predicted value $\hat{f}_t$. Such modeling is unnatural. For example, if we set $\psi = g(x)^2+f_s (x)$, Assumption 2.2 does not hold. In this case, we get the same predicted value even if the sign of $\hat{g}(x)$ is reversed.
> In the revised manuscript, we will add the above explanation to clarify that Assumption 2.2 is never restrictive.
>
>
> > **[Q1]** In the formulation of sec.2.1, it seems that $f_t (x)$ and $f_s (x)$ have the same dimension of $y$, i.e. they both belong to $\mathbb{R}$. However, in the examples below $y$ has the same dimension as the scalar product $\langle \theta, f_s (x) \rangle$. Can you clarify?
>
> In our setup, $f_t (x)$ is a scalar, while $f_s (x)$ is a vector. Since the current notation is misleading, we will revise the manuscript to represent vectors in bold.
>
> **[Minor comments]**
> Thank you very much for pointing out our typos. We will fix the errors.
>
> [References]
> S. S. Du, et al. “Hypothesis transfer learning via transformation functions,” NeurIPS, 2017.

---

> > ### Comment · Reviewer_DK8u · 2023-08-15
> >
> > Thank you for your rebuttal and clarifications.
> >
> > [RW1] Yes, this would have helped. I think this will improve clarity. Also, a description of the goal of the 3 steps in the TL procedure could help.
> >
> > Personally, I keep finding the paper hard to parse in section 2.1. Coming back to the paper after a few weeks I needed time to reconcile the various definition.
> >
> > On the other hand, the context of this work is new to me, and I may find confusing this community's jargon and notation.

---

> > > ### Author Response · Authors · 2023-08-17
> > >
> > > We thank the reviewer for the careful reading of our manuscript. We are pleased to hear that your concerns have been partially addressed.
> > >
> > > > Also, a description of the goal of the 3 steps in the TL procedure could help.
> > >
> > > To better clarify the purpose of the three-step procedure, we will add the following sentences after line 80 on page 2:
> > >
> > > The objective of step 1 is to identify a transformation $\phi$ that maps the output variable $y$ to the intermediate variable $z$, making the variable suitable for learning. In step 2, a predictive model for $z$ is constructed. Since the data is limited in many TL setups, a simple model, such as a linear model, should be used as $g$. Step 3 is to transform the intermediate model $g$ into a predictive model $f_t$ for the original output $y$.

---

> > > > ### Comment · Reviewer_DK8u · 2023-08-17
> > > >
> > > > Thank you. I update my score.

---

### Decision · Program_Chairs · 2023-09-21

**Decision:**

Accept (poster)

**Comment:**

The recommendation is based on the reviewers' comments, the area chair's personal evaluation, and the post-rebuttal discussion.

This paper provides new theoretical understandings and practical insights into transfer learning in the considered affine model transformation setting. Most reviewers find the results sufficiently novel and convincing. The authors’ rebuttal has successfully addressed the major concerns of reviewers. Therefore, I recommend acceptance of this submission. I also expect the authors to include the new results and discussion during the rebuttal phase to the final version.